



# A field study on ice melting and breakup in a boreal lake, Pääjärvi, in Finland

Yaodan Zhang[1,2], Marta Fregona[3], John Loehr[2], Joonatan Ala-Könni[4], Shuang Song[5,6], and Matti Leppäranta[2], and Zhijun Li[1]

[1]State Key Laboratory of Coastal and Offshore Engineering, Dalian University of Technology, Dalian, China
[2]Lammi Biological Station, University of Helsinki, Finland
[3]Department of Civil, Environmental and Mechanical Engineering, University of Trento, Italy
[4]Institute of Atmospheric and Earth Sciences, University of Helsinki, Helsinki, Finland
[5]Water Conservancy and Civil Engineering College, Inner-Mongolia Agricultural University, Hohhot, China
[6]College of Water Conservancy, Shenyang Agricultural University, Shenyang, China

*Correspondence to*: Zhijun Li (lizhijun@dlut.edu.cn), Yaodan Zhang (zhangyaodan@mail.dlut.edu.cn).

**Abstract.** Lake ice melting and breakup form a fast, nonlinear process with important mechanical, chemical, and biological consequences. The process is difficult to study in the field due to safety issues, and therefore relatively little is known about its details. In the present work, ice monitoring was based on foot, hydrocopter, and boat to get a full time-series of the evolution of ice structure and geochemical properties through the melting period. The field observations were made in Lake Pääjärvi during the ice decay periods in 2018 and 2022. In 2022, the maximum thickness of ice was 55 cm with 60 % snow-ice, and based on the data and heat budget analysis, the ice melted by 33 cm from the surface and 22 cm from the bottom while porosity increased to 40–50 % at breakup. In 2018, the snow-ice layer was small and bottom and internal melting dominated during the decay. Due to global warming, the ice breakup date became earlier. The mean melting rates were 1.31 cm d$^{-1}$ in 2022 and 1.55 cm d$^{-1}$ in 2018. In 2022 the electrical conductivity (EC) in ice was 11.4±5.79 S cm$^{-1}$, one order of magnitude lower than in the lake water, and ice pH was 6.44±0.28, lower by 0.4 than in water. pH and EC of ice and lake water decreased along the ice decay except slight increases in ice due to flushing by lake water. Chlorophyll *a* was less than 0.5 g L$^{-1}$ in porous ice, approximately one-third of that in the lake water. These results are important for further development of numerical models and understanding the process of ice decay with consequences to lake ecology and to safety of ice cover for human activities.



## 1 Introduction

Lake ice is a thin layer between the atmosphere and lake water and plays an important role in the meteorological, hydrological, biological, geochemical and socio-economical regimes of boreal lakes (Leppäranta, 2015). Lake ice affects the local weather altering the heat, mass and momentum exchange between the atmosphere and water bodies and increase the albedo, reducing the solar radiation transfer into the water (Ellis and Johnson, 2004; Rouse et al., 2008a, 2008b; Williams et al., 2004). The physical properties of ice cover are determined by stratification, crystal structure, gas bubbles and porosity. These properties to a large degree control ice mechanics, acoustics, optics, thermodynamics and electrodynamics which have a key role in ice remote sensing, the living conditions under-ice, and the ecology within ice (Iliescu and Baker, 2007; Li et al. 2010; Shoshany et al., 2002). Although most boreal lakes possess a seasonal ice cover, lake research has traditionally focused on summer, and especially little is known about the decay of ice when the ice starts to melt and weaken. The obvious reason is that at this time fieldwork is logistically very difficult to carry out. However, the physical and geochemical properties of ice undergo rapid changes during the ice decay period that has an important influence on conditons on and below the ice cover.

There are two major practical problems with melting lake ice due to loss of ice strength caused by the deterioration of ice (Ashton, 1985; Leppäranta, 2015; Masterson, 2009). The bearing capacity of ice decreases, and therefore on-ice traffic becomes risky. Accidents are reported every spring due to ice breakage, connected with fishing or crossing of lakes. The variations of ice structure during the ice decay period seriously impact the form and time of ice breakup in the spring. Decreasing ice strength implies that ice cover may be broken by wind and drift on shore. Also, moving ice with finite strength is a risk for hydraulic structure, such as lake site platforms, bridges and a force for near-shore erosion. Hence, it is urgent to study the physical properties of ice during melting period.

The climatology of ice breakup date has been widely studied based on long-term time-series records (Benson et al., 2012; Korhonen, 2006; Karetnikov et al., 2017; Magnuson et al., 2000). A steady trend toward earlier melting date has been reported in most recent ice phenology studies, by about one week over 100 years and can be attributed to the global climate warming. Some numerical modelling studies of ice breakup date revealed that the time when ice starts to melt and the internal deterioration has





important impact on the accuracy of simulations (Yang et al, 2012). The physics of climate sensitivity
and the relationship to the timing of ice breakup is a question of atmospheric warming and falling
albedo (Leppäranta, 2014). Understanding better this phenological change requires a quantification of
the physical mechanisms that control the melting of ice.
The trend for earlier melting of lake ice is considered to be the driving factor for the changes of
ecological and biogeochemical processes in seasonal ice-covered lakes (Garcia et al, 2019; Griffiths et
al, 2017). Lake ice interacts with under ice water to further drive or facilitate the migration and
transformation of nutrients and metals, resulting in changes in the biomass and structure of
phytoplankton (Cavaliere and Baulch, 2018; Schroth et al, 2015). In addition, the habitat conditions and
ecosystem structure under the ice affect the limnology of the following seasons (Hampton et al., 2017).
pH, Electrical conductivity (EC) and Chlorophyll *a* (Chl *a*) are important indicators of ecological
environment and have significant impacts on the primary productive. However, it is uncommon to see
pH, EC and Chl *a* quantified during ice decay period. In general, an overall lack of knowledge of the
extent to how ice melting affects ecological and biogechemical process limits the properly assess the
impacts of climate change on limnological process in cold regions (Tan et al., 2018).
In the period of ice cover decay, the snow layer melts first. Due to its low light transmissivity, the snow
cover protects the ice by its presence (Ashton, 1986; Leppäranta, 2015; Warren, 1982). Also, the high
albedo of snow delays the start of the ice decay period. The situation changes immediately when the
snow melting begins, and the snow cover disintegrates. Then ice melting begins, and also sunlight
penetrates the ice to heat the water under the ice depending on the spring weather and ice quality
(Kirillin et al. 2012). At the same time, primary production begins and as the ice melts, all impurities
contained in the ice are released into the water or to the air which may change the water environment.
Normally primary production peaks after ice breakup; thus, ice melting is connected to the spring bloom.
Due to the difficult conditions with unstable and deteriorating ice cover, there has not been much in situ
research during the ice melting period. Knowledge of melt rate is limited to a few studies, with typically
$1-3\mathrm{cm}\ \mathrm{d}^{-1}$ in terms of equivalent ice thickness, seen at the top and bottom boundaries and in the ice
interior, depending on the weather conditions (Jakkila et al., 2009; Leppäranta et al., 2010, 2019; Wang
et al., 2005; Yang et al., 2012). Surface melting is mainly related to the albedo. It was found that the





transmittance changed with the internal melting and the amount of gas pockets and water-filled pockets
in ice (Jakkila et al., 2009). Internal melting opens channels for flushing the ice by surface melt water
and lake water. It is mainly reflected in the increase of porosity. When the porosity of ice has reached
the level of around 0.5, ice cover collapses by its own weight and then disappears rapidly (Leppäranta et
al., 2010, 2019). Bottom melting is caused by the heat flux from water that can be large in spring, and in
the cold season this heat flux provides a limitation for the ice growth (Shirasawa et al., 2006; Yang et al.,

90 2012).

We examine here the decay of ice cover in Lake Pääjärvi, southern Finland by field surveys and ice and
water samples in two years, April 2018 and 2022. This lake is frozen for 4–5 months annually, and the
ice cover consists of congelation ice and snow-ice with snow cover on top (Jakkila et al., 2009; Wang et
al., 2005). The decay of ice cover takes about one month, and the process is controlled by the presence
of snow on top and the optical quality of snow, in addition to atmospheric and solar forcing. The
structure and properties of the ice are changing during the decay process, and the actual melting of the
ice takes place at the surface and bottom and in the interior. This paper gives the final results of the field
campaigns.

## 2 Materials and methods

### 2.1 Study site

Lake Pääjärvi is located in the boreal zone in southern Finland (61°40′ N, 25°08′ E). The lake area is
13.4 km$^2$, the mean and maximum depths are 14.4 m and 87 m, respectively, and the catchment area is
244 km$^2$ (Arvola et al., 1996). Lake Pääjärvi is a humic, brown-water lake with an average optical depth
of 0.67 m and Secchi depth of 1.8 m (Arst et al., 2008). The ice season lasts normally 4–5 months. In
the period 1910–1988, the mean freezing and breakup dates were December 13 and May 5, respectively.
For the breakup date the standard deviation was 8 days, the earliest and latest dates were April 14 and
May 18, respectively, and the maximum annual ice thickness was 50 cm with standard deviation of 9
cm (Kärkäs, 2000). The fraction of snow ice was on average one-third in 1993–1999 (Leppäranta and
Kosloff, 2000).





The field study was made in Pappilanlahti Bay in the west side of the lake. This bay is shallow
(maximum depth <15 m), with three small inflows at the end of the bay and a weak groundwater flux at
the bottom. There was access to the lake ice from a platform at the shore by foot and in late season by a
hydrocopter and a boat. Our field observations were made as a pilot study in 12–20 April 2018 and as
the main experiment in 25 March–3 May 2022, of which the latter case was more extensive and thus
provides the body of the data. The ice situation was recorded by ground and drone orthophotos and field
notes, and ice and water samples were collected several times. In 2022 the whole decay period was
mapped while in 2018 just the last eight days of it.
**2.2 Observations**
In the pilot study in 2018, the field site was visited five times between April 12 and April 20. The study
was focused on a short period at the end of ice decay. On April 12, 15 and 20 ice samples were taken.
After April 20, because of the rapid melting, it was not possible to walk on the ice or to use a boat for
sampling, but photographs were taken daily from the shore. Otherwise sampling work was done in
similar manner as in 2022.
In 2022 the monitoring took 40 days on a weekly basis. Each time the ice quality and thickness were
checked first. Ice samples (whose lateral cross-section was 30 cm × 30 cm) were cut by drill and saw
and stored then in a freezer. Water samples were taken from the drill holes and analysed in the
laboratory for pH, EC and Chl *a*. The ice samples were analysed in a cold room (–10 °C) for the crystal
structure and density. Ice melt water was also analysed for the pH, EC and Chl *a* the same way as the
water samples.
The study period in 2022 covered the whole decay process. Eight field site visits were made from
March 25 to May 3. The sampling was made by foot from shore until April 22. Since April 26, melting
begins from the shoreline and there was a slush layer between the surface ice and the congelation ice
layer, the bearing capacity of the ice was not strong enough for walking on the ice at the latest phase of
the melting period. Then, a hydrocopter was used for ice sampling on April 26–29 (Fig. 1a). On May 3,
the melting created several open channels, a boat was used for ice sampling. The freeboard, snow




thickness, snow-ice thickness, and congelation ice thickness were measured by ruler during the ice

sampling and the water samples were collected after the ice sampling into a sealed bottle.

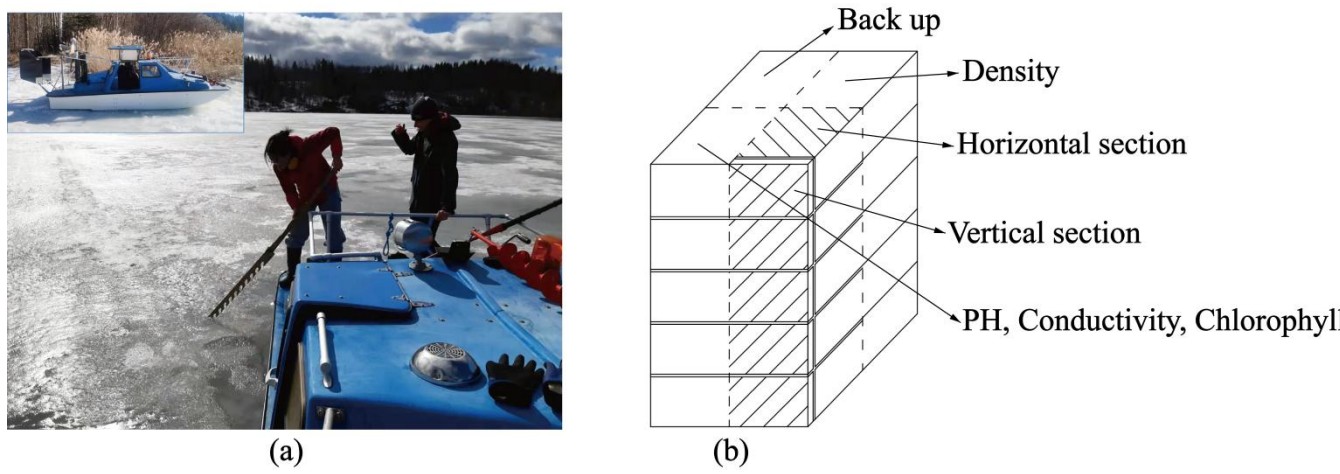

**Figure 1. Lake ice sampling and processing: (a) collect ice with a handsaw on the hydrocopter; (b) the ice block was sliced into four parts for different observations.**

All ice and water samples were put into plastic bags at the site and transported immediately to Lammi

Biological Station (about 500 m away from the site). Then, the ice samples were stored in a freezer at a

temperature of −18 ℃, and the water samples were stored in fridge at a temperature of 4–6 ℃. In the

analysis, each ice sample was divided into four sections. Section 1 was cut vertically into layers, and

then the pH, EC and Chl $a$ of the layers was measured from the meltwater. Section 2 was cut vertically

and horizontally to map the ice crystal structure and study the gas bubbles by image analysis, Section 3

was used to measure the density of ice, and Section 4 was stored as a backup (Fig. 1b).

Available routine meteorological and hydrological data of the Finnish Meteorological Institute (FMI)

and Finnish Environment Institute (SYKE) were utilized. SYKE data include manual measurements of

thicknesses of ice, snow-ice and snow, and freeboard every ten days during the whole winter in

Pappilanlahti Bay, and FMI provided the meterological data of an automated station in the yard of the

Lammi Biological Station half a kilometre from our site. The SYKE data was used for the all-season ice

and snow thickness, while the melting period data were own field observations. The data base of the

Lammi Biological Station was utilized for the long-term ice phenology and geochemistry of inflows

from brooks into the study bay.





## 2.3 Laboratory work

The ice crystal structure, gas bubbles, and ice density were studied from the ice samples in the INAR (Institute of Atmospheric and Earth Sciences, University of Helsinki) ice laboratory. The crystal structure was obtained from thin sections. The samples were cut into vertical sections of 8–10 cm height by a bandsaw, and horizontal sections were extracted at the vertical cuts. The sections were frozen on glass plates to be prepared for thin sections. The size and distribution of gas bubbles in the ice were observed under normal light, and ice crystal structure was obtained from thin sections between crossed polarizers (Deng et al., 2019; Langway, 1958).

Measurements of ice density can be found in several studies (Timco and Frederking, 1996). The mass/volume method was used to measure the ice density in laboratory, and the freeboard in the field was used as a control. In the laboratory, the sample was cut into 5 cm cuboids by a bandsaw. The sides of a cuboid were measured by vernier caliper, and the mass was measured by an electronic scale with the accuracy of 0.001 g.

For the geochemistry, the samples were cut into vertical sections based on the structure at an interval of 8–10 cm by a bandsaw. Then, the blocks were melted in sealed bags, the water was poured into sample bottles and stored in a fridge (at 4–6℃). pH and EC were measured from unifiltered samples according to the standard in SFS-EN 27888 and SFS 3021. By using a Thermo Orion 3-STAR Precision Benchtop pH meter and YSI 3200 conductivity sensor, respectively. These two instruments offer high accuracy for water analyses and multipoint calibration. The amount of the Chl $a$ was measured from filtered sub-samples by Shimatzu UV-1800 spectrophotometer (Arvola et al. 2014). The absorbance of Chl $a$ was extracted at a long wavelength.

## 3 Results

### 3.1 Ice structure

The ice decay period began on March 25 and the final breakup took place on May 5, 2022 (Table 1). The thickness of ice was 55 cm on March 25. The ice was melting at both boundaries and in the interior. On April 22, it was still possible to walk on ice when the total thickness was 38 cm but the ice was quite



porous. The decay period was 42 days. The melting rate increased toward the end, and the mean value
was 1.31 cm d$^{-1}$.
**Table 1. Thickness of ice layers and freeboard in the melting phase (cm) and porosity (%) in 2022, also shown is the**
**ratio of freeboard to draft.**

| 2022 | Snow-ice | Congelation ice | Total ice | Porosity | Freeboard | Fb/draft | Snow |
|---|---|---|---|---|---|---|---|
| March 25 | 33 | 22 | 55 | x | 5.5 | 0.11 | 1 |
| April 1 | 31 | 20 | 51 | 6.1 | 5 | 0.11 | 2.5 |
| April 8 | 30 | 17 | 47 | x | 2 | 0.044 | 13 |
| April 14 | 31 | 17 | 48 | 7.7 | 5 | 0.12 | 2 |
| April 22 | 27 | 11 | 38 | 15.2 | 4 | 0.12 | 0 |
| April 26 | 7.5 + 7¶ | 10 | 24.5 | 17.1 | 1 | 0.0057 | 0 |
| April 29 | 6 + 12¶ | 4 | 22 | 24.1 | 0.5 | 0.0023 | 0 |
| May 3 | z | 2–z | 2 | 34.0 | x | x | 0 |
| May 5 | z | 2–z | 0 | 0 | 0 | x | 0 |

¶ Surface ice + slush layer
Generally, when deterioration is occurring the ice cover has a grayish, splotchy appearance from above
and appears treacherous (Ashton, 1985). Figure 2 shows images of the ice cover recorded by drone
orthphotos at an altitude of 100 m during the melting period. The snow fell at the beginning of April
turned the ice white. As the air temperature rose, the snow on the ice began to melt, creating a patchy
surface that deteriorated until the ice broke up.

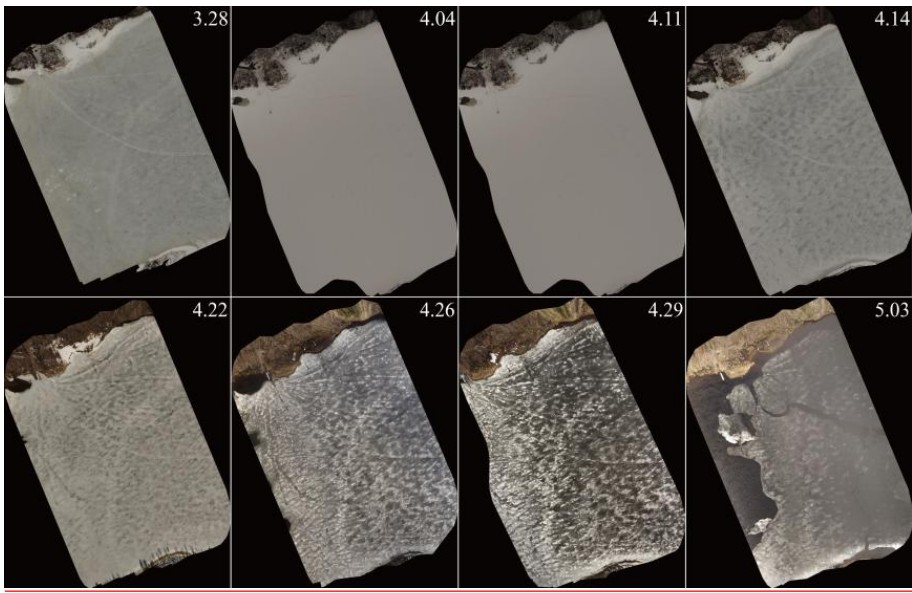


**Figure 2. Drone orthophotos of the ice cover in the melting period (time given as month.day).**



The melting period began in late March. The maximum annual ice thickness of 55 cm was well within
the range of long-term statistics where the mean value is close to half meter (Kärkäs, 2000). As we can
see from Fig. 3a–f, there were two principle vertical layers in the Lake Pääjärvi ice cover. The top layer
was granular snow-ice, the grain size was 1–9 mm with blurred crystal boundaries, and the lower layer
was columnar congelation ice. The columnar ice layer was clear ice with the grain size of 2–10 cm.
With the increasing air temperature, the ice crystal structure results showed that the thickness of both
snow-ice and congelation ice decreased, and the porosity became more and more.
The ice melted 4 cm in May 25–April 1 (0.57 cm d$^{-1}$).  On April 1, it was seen from the crystal structure
that the shape of snow-ice crystals above 28 cm was very irregular with blurred crystal boundaries, and
the grain size was mainly within 1–2 mm. The grain size of the 28–32 cm layer was mainly within 2–5
mm, granular crystals with clear boundaries. It can be judged that the top 0–28 cm layer was snow-ice
that had undergone the thawing-refreezing process, and the 28–32 cm layer was the surface congelation
ice layer formed at the beginning of the ice season. The columnar ice layer underneath was clear ice
with grain size increasing with depth, range from 2 to 10 cm. There was a volume of 4–6 % rachis
shaped and spherical shaped gas bubbles in snow-ice with the maximum diameter of 4 mm, and a
volume of 1–2 % spherical shaped gas bubbles in congelation ice with the maximum diameter of 1 mm.
From the vertical sections, there was also a distinct boundary between granular ice and columnar ice at
around 32 cm.
Then, in April 1–14 the melting was 4 cm (0.30 cm d$^{-1}$), but the thickness of snow-ice was unchanged.
According to the weather data, continuous snowfall began on April 5, and the temperature rose after
that, resulting in the formation of new snow-ice through the melt-freeze cycle. Compared with April 1,
the ice crystal size had not changed, but the temperature rose from April 10 to 14. The bubble content in
the snow-ice was 5–7 %.
After April 14, the temperature continued to rise, and the ice rapidly melted, 10 cm in April 14–22 (1.25
cm d$^{-1}$). The horizontal and vertical sections showed that severe melting occurred at the snow-ice grain
boundaries. The gas content in snow-ice increased to 6–10 % and 1–3 % in congelation ice. Also, the
maximum diameter of gas bubbles increased to 5 mm in snow-ice and 3 mm in congelation ice.



In April 26−29 (0.83 cm d$^{-1}$), a slush layer appeared below a surface ice layer due internal melting of
ice. Since April 26, the columnar ice began to melt at crystal boundaries, and gas inclusions appeared at
the boundaries. On April 29, gas bubbles also appeared in the inside columnar crystals, with the bubble
content reaching 5 % and the maximum bubble size reaching 5 mm. On May 3, the columnar ice and
slush layers had melted, and 2 cm snow-ice left.

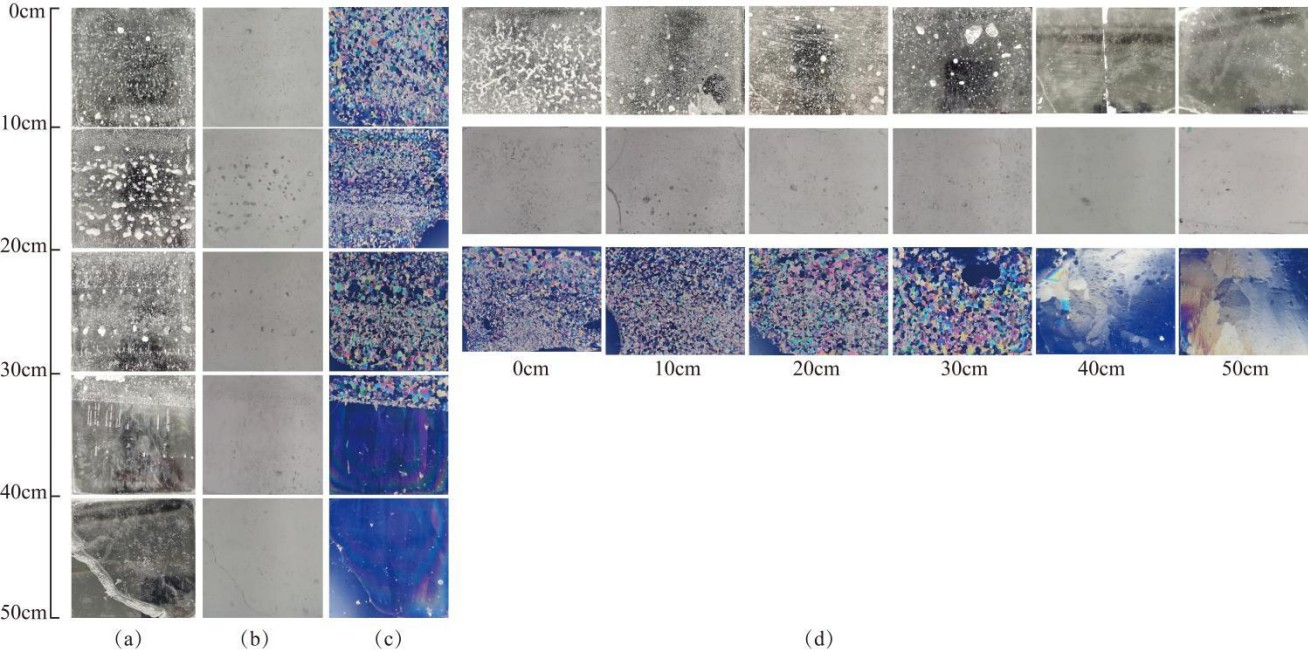


**Figure 3a. Lake Pääjärvi ice crystal structure of April 1.**


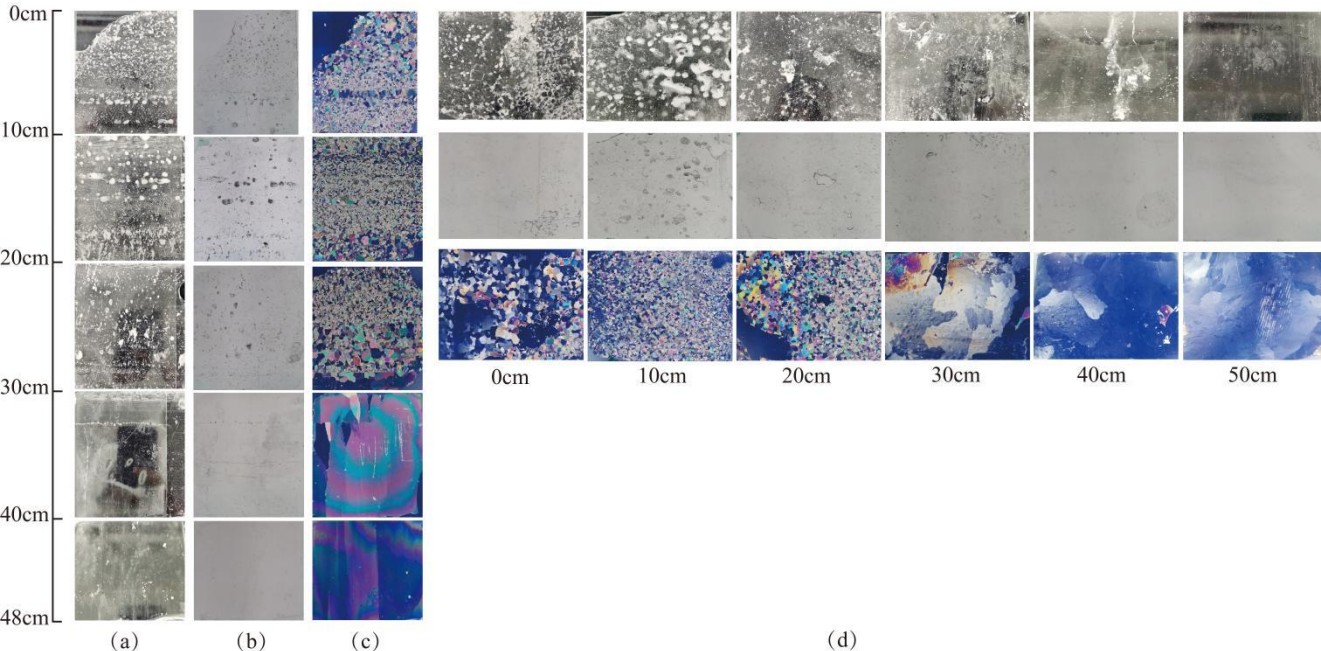

**Figure 3b. Lake Pääjärvi ice crystal structure of April 14.**

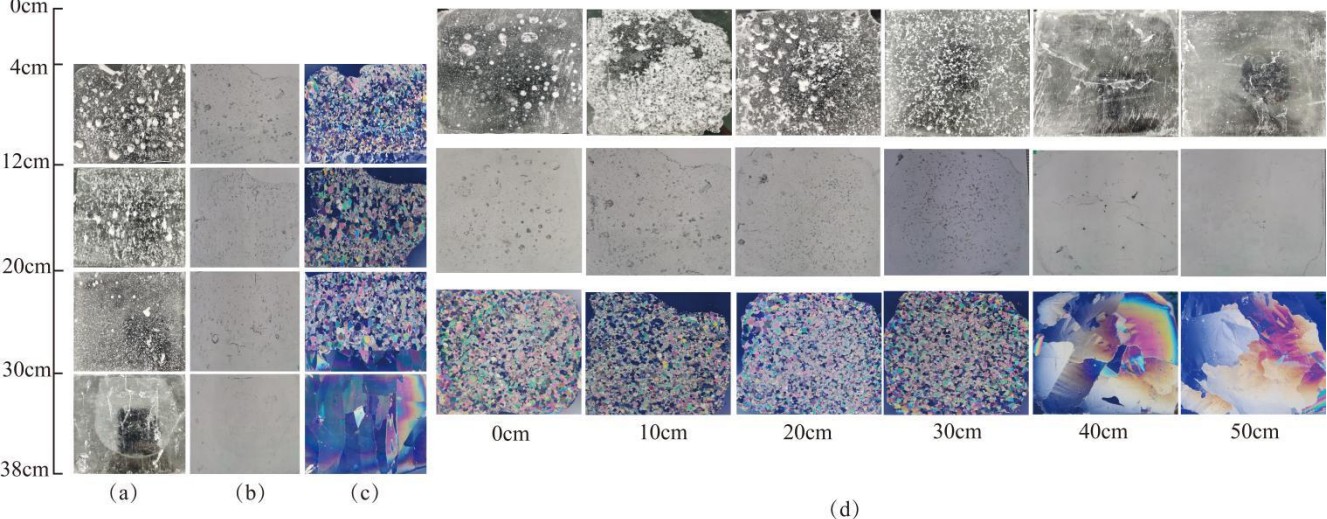

**Figure 3c. Lake Pääjärvi ice crystal structure of April 22.**




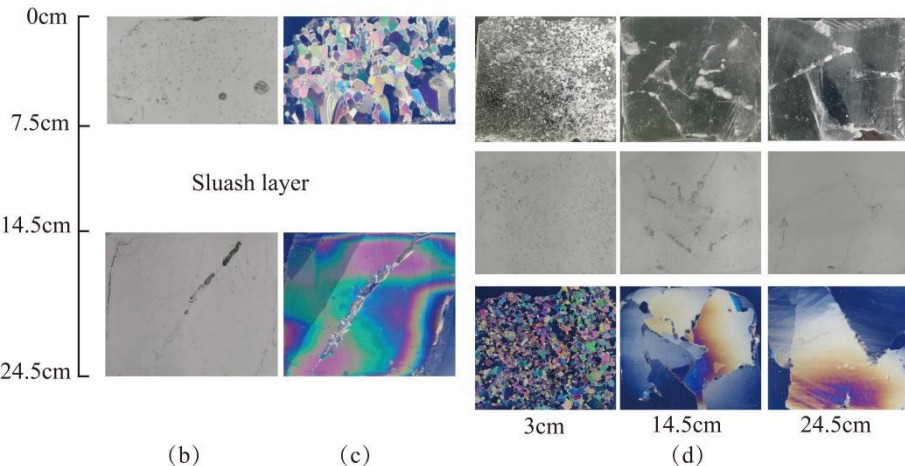

**Figure 3d. Lake Pääjärvi ice crystal structure of April 26.**

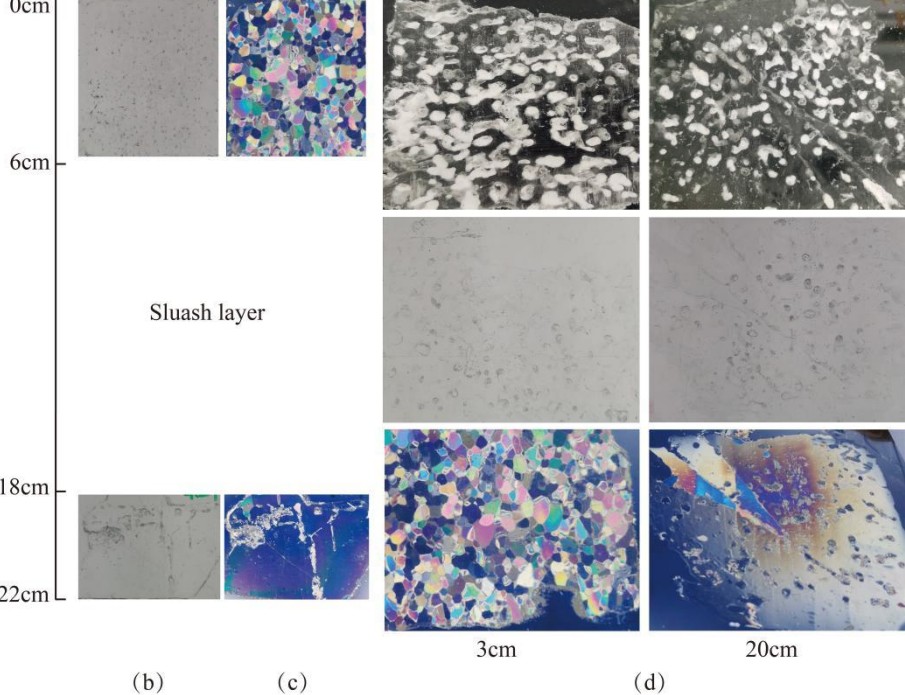

**Figure 3e. Lake Pääjärvi ice crystal structure of April 29.**



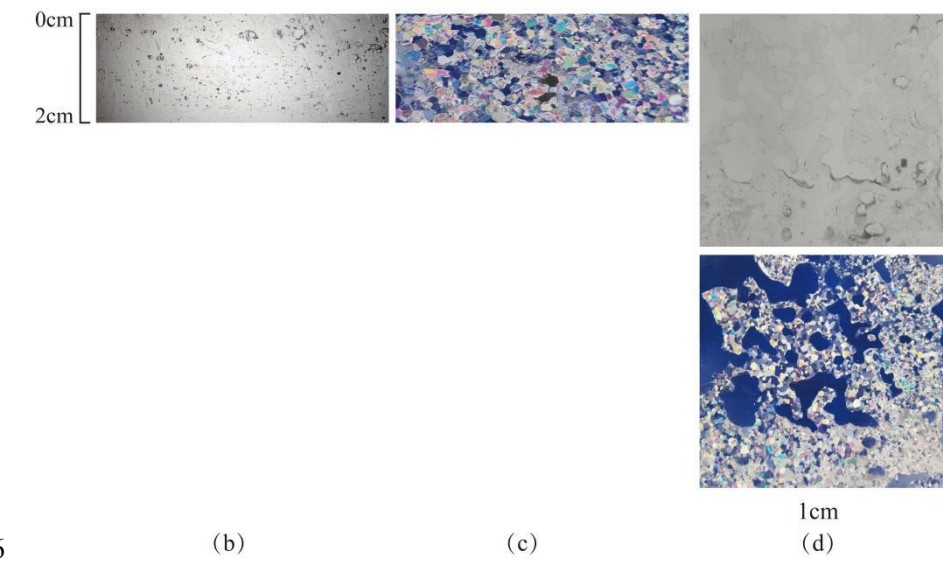

236           (b)         (c)         (d)

**Figure 3f. Lake Pääjärvi ice crystal structure of May 3.**
**Figure 3. Lake Pääjärvi ice crystal structure between March 25 and May 3.(a) photographs of gas bubbles with the**
**thickness of the vertical cross-section around 5mm in normal light; (b) photographs of gas bubbles with the thickness**
**of the vertical cross-section around 1mm in normal light; (c) photographs of the vertical cross-section of the crystal**
**structure in polarized light; (d) photographs of the horizontal cross-section of gas bubbles and crystal structure.**
**Photographs of gas bubbles with the thickness of the vertical cross-section around 5mm in normal light were missing**
**on April 26, April 29 and May 3.**
In 2018, the decay period also began at the end of March, and the final breakup took place on April 25.
The thickness of ice was 42 cm on March 30. The ice was melting by 0.5 cm d$^{-1}$ at the bottom, and on
April 2 a 14 cm new snow layer fell and then melted in 10 days. On April 12 the ice was bare and solid,
and ice thickness was 35.0 cm, consisting of 5.3 cm snow-ice and 29.7 cm congelation ice. In April 5–
10, the average daily air temperature was above 0 °C, but in April 10–15 it was below 0 °C in the night
time. It was raining on April 1, 3, 8, 19, and on the 24th the rain greatly accelerated the melting. After
April 12, the thickness of ice started to decrease along with the rising air temperature and solar radiation
(Table 2). The ice melted 4 cm in April 12–15, in April 15–20 the melting was 12.7 cm, and by April 20
the 5.3 cm snow-ice layer had melted fully while congelation ice thickness had decreased by 9.4 cm
with 20.3 cm remaining. Between 12–20 April, it was possible to walk on the ice from the shore. In all,
the ice decay period lasted 27 days, and the mean melt rate was 1.55 cm d$^{-1}$.
**Table 2. Thickness of ice layers and freeboard in the melting phase (cm) and porosity (%) in April 2018, also shown is**
**the ratio of freeboard to draft.**



| 2018 | Snow-ice | Congelation ice | Total ice | Porosity | Freeboard | Fb/draft |
|------|----------|-----------------|-----------|----------|-----------|----------|
| April 12 | 5.3 | 29.7 | 35.0 | ~ 0 | 3.0 | 0.094 |
| April 13 | 4.7 | 29.3 | 34.0 | x | 3.0 | 0.097 |
| April 14 | 3.3 | 28.7 | 32.0 | x | 2.0 | 0.067 |
| April 15 | 2.7 | 28.3 | 31.0 | x | 2.0 | 0.069 |
| April 20 | 0 | 20.3 | 20.3 | 25 | x | x |
| April 25 | 0 | 0 | 0 | x | 0 | x |

The ice sample data in Tables 1−2 were used to determine the melting at the surface (snow-ice) and
bottom (congelation ice), and the porosity was used to estimate internal melting. The result for 2022
(Table 3) shows that the snow-ice melted from the top and congelation ice from the bottom almost fully,
and the last 2 cm piece was snow-ice. The mean melt rate at the bottom was 0.38 cm d⁻¹ in March 25 –
April 26 that corresponds to the energy flux of
$$\frac{h_f}{h_d} = \frac{\rho_w - \rho_d}{\rho_f} = 13 \text{ W m}^{-2},$$  (1)
where $\rho_i$ is ice density, $L_f$ is the latent heat of freezing and the time is $\Delta t = 1$ d. The energy flux was a
little larger than normally assumed. The internal melt rate was 0.18 cm d⁻¹ equivalent thickness that was
limited due to the low transmittance of snow-ice. In the last week of existence the structure of ice was
highly porous and internal breakages occurred.
**Table 3. Ice melting in spring 2022 (cm). The numbers show the change from the row above to the present one.**

| 2022 | Surface melt | Bottom melt | Total melt | Internal melt |
|------|--------------|-------------|------------|---------------|
| March 25 | 0 | 0 | 0 | 0 |
| April 1 | 2 | 2 | 4 | x |
| April 8 | 1 | 3 | 4 | 0.4 |
| April 14 | -1 | 0 | -1 | 0.4 |
| April 22 | 4 | 6 | 10 | 3.2 |
| April 26 | 4.5 | 1 | 5.5 | 0.6 |
| April 29 | 4.5 | 6 | 10.5 | 1.6 |
| May 3 | 16 | 4 | 20 | 1.2 |
| May 5 | 2 | 0 | 2 | 0 |
| Sum | 33 | 22 | 55 | 7.4 |

Winter 2018 ice cover was different from 2022 in that the ice was mostly (85 %) congelation ice. Table
4 shows that the snow-ice had all melted by April 20 when there was still 20.3 cm congelation ice left.
In 12−20 April the surface melting was 5.3 cm, the bottom melting was 9.4 cm, and internal melting



was 6.9 cm. The ice was more transparent that allowed more sunlight penetration through ice than in 2022. The bottom melting in 12–15 April corresponds to the heat flux of 16 W m$^{-2}$ from water to ice.

**Table 4. Ice melting in spring 2018 (cm). The numbers show the change from the row above to the present one.**

| 2018 | Surface melt | Bottom melt | Total melt | Internal melt |
|---|---|---|---|---|
| April 12 | 0 | 0 | 0 | 0 |
| April 13 | 0.6 | 0.4 | 1.0 | x |
| April 14 | 1.4 | 0.6 | 2.0 | x |
| April 15 | 0.6 | 0.4 | 1.0 | x |
| April 20 | 2.7 | 8.0 | 10.7 | 6.9 |
| April 25 | z | 20.3–z | 20.3 | x |

## 3.2 Ice density

At the initial stage of melting, in April 1–14, 2022, the average densities of snow-ice and congelation ice were 850 kg m$^{-3}$ and 970 kg m$^{-3}$, respectively. Since April 22, no new snow-ice was formed and the ice continued melting at the surface, bottom and in the interior. Accordingly, the ice density profiles were moved along the direction of ice depth, as shown in Fig. 4, and the depth of the movement was consistent with the ice melting thickness from the surface. In the melting process, the density of snow-ice and congelation ice decreased gradually, with density higher with depth. In particular, on April 22 the snow-ice density increased greatly with depth. The pore channels in the ice did not penetrate into water, and internal melting may have caused meltwater to accumulate in some parts of the ice, resulting in large ice density, even more than the density of ice on April 14. The average density of snow-ice was 730 kg m$^{-3}$ and the average density of congelation ice was 930 kg m$^{-3}$ on April 22. Finally, from April 26 to May 3, the average density of snow-ice and congelation decreased to 690 kg m$^{-3}$ and 770 kg m$^{-3}$, respectively. The density data were used to estimate the porosity, which was found to increase from 6.1 % to 34 % during the melting season (Table 1).





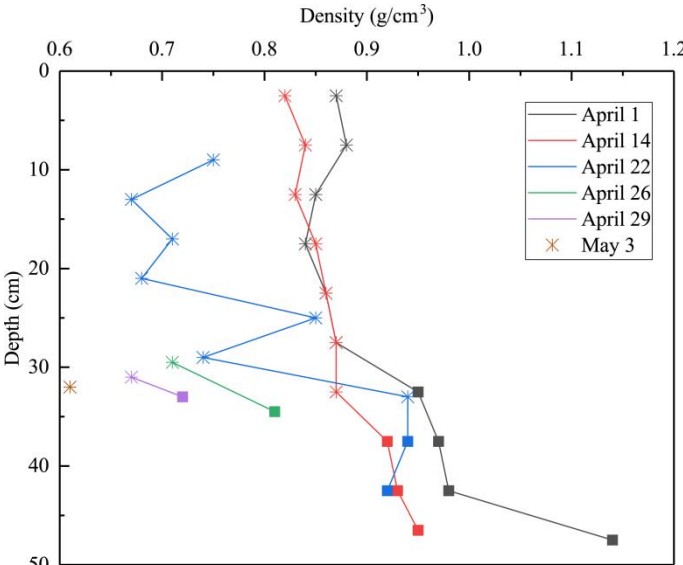

**Figure 4. Lake Pääjärvi ice density profiles (asterisk stands for snow ice, cube for congelation ice).**
For bare ice, the freeboard/draft ratio is
$$\frac{h_f}{h_d} = \frac{\rho_w - \rho_d}{\rho_f}, \tag{2}$$
where $h$ is total ice thickness, $\rho$ is density, and the subscripts are for water, $d$ for draft, and $f$ for
freeboard. In winter, for $\rho_f = \rho_d \approx 910$ kg m$^{-3}$, this ratio is 0.099 or $^1/_{10}$. It increases when the porosity
decreases, but it may decrease if meltwater drainage from freeboard is trapped inside the draft to reduce
the buoyancy. This is consistent with field observation in 2022 and 2018. In practice it is difficult to
determine the freeboard/draft ratio as it requires an order of one-millimetre accuracy for the freeboard.
**3.3 Ice geochemistry**
During the melting period, meltwater was mixed into the surface layer below the ice and influenced the
water chemistry. The meltwater had lower pH and EC than the lake water (Table 5), and consequently
lower density (Kirillin et al., 2012), and therefore a thin fresh surface layer could form just under the ice.
In the winter of 2021–2022, before the snowfall on April 5, the pH and EC of snow-ice decreased. The
mean pH of snow-ice was 6.47 on March 25 and 6.38 on April 1, and in these dates the mean EC were
23.0 S cm$^{-1}$ and 17.3 S cm$^{-1}$, respectively. Then EC decreased with average value of 9.34 S cm$^{-1}$ on
April 14 still slightly decreasing thereafter. In congelation ice, EC was consistently within 8–11 S cm$^{-1}$.





With the process of ice decay, melting in upper layer of the ice cover drained down into the lower layer
of the ice cover which caused the higher EC of April 1 20–31 cm, April 8 30–40 cm and April 14 42–48
cm. Until April 22, pH was smaller in in snow-ice than in congelation ice but EC were greater in snow-
ice than in congelation ice. After the slush layer was created on April 26, pH, EC and Chl $a$ were
slightly higher in the bottom congelation ice than in snow-ice due to the flooding of the lake water. The
chlorophyll $a$ content was greater in snow-ice than in congelation ice but less than that in lake water
before April 22.
In the winter of 2017–2018, EC was stable at 97 S cm$^{-1}$ under ice until dropping to 81 S cm$^{-1}$ on April
20th. pH beneath the ice also decreased slightly in the progress of melting, from 6.87 to 6.77. In ice
meltwater, EC was 6 S cm$^{-1}$ and pH was 6.35.
**Table 5. pH, EC and Chl $a$ in ice meltwater and water under ice at the study site in 2022 and 2018.**

| Year | Date | Depth (cm) | Ice type | Ice | | | Under ice | | |
|------|------|-----------|----------|-----|----|----|----|----|----|
| | | | | pH | EC (µS cm$^{-1}$) | Chl $a$ (µg L$^{-1}$) | pH | EC (µS cm$^{-1}$) | Chl $a$ (µg L$^{-1}$) |
| 2022 | March 25 | 0–10 | Snow-ice | 6.47 | 31.1 | 0.3 | | | |
| | | 10–20 | Snow-ice | 6.46 | 24.3 | 0.3 | | | |
| | | 20–30 | Snow-ice | 6.46 | 13.6 | 0.6 | x | x | x |
| | | 30–40 | Cong.ice | 6.75 | 8.95 | 0.4 | | | |
| | | 40–50 | Cong.ice | 6.75 | 8.94 | < 0.1 | | | |
| | April 1 | 0–10 | Snow-ice | 6.39 | 16.8 | 0.2 | | | |
| | | 10–20 | Snow-ice | 6.38 | 14.0 | 0.2 | | | |
| | | 20–31 | Snow-ice | 6.36 | 21.3 | 0.6 | x | x | x |
| | | 31–40 | Cong.ice | 6.69 | 8.54 | < 0.1 | | | |
| | | 40–50 | Cong.ice | 6.69 | 8.53 | 0.1 | | | |
| | April 8 | 0–10 | Snow-ice | 6.35 | 9.66 | 0.2 | | | |
| | | 10–20 | Snow-ice | 6.35 | 8.17 | 0.5 | | | |
| | | 20–30 | Snow-ice | 6.34 | 10.7 | 0.6 | x | x | x |
| | | 30–40 | Mix-ice | 6.62 | 16.3 | 0.2 | | | |
| | | 40–50 | Cong.ice | 6.64 | 8.51 | 0.3 | | | |
| | April 14 | 0–11 | Snow-ice | 6.28 | 10.8 | 0.4 | | | |
| | | 11–22 | Snow-ice | 6.28 | 8.92 | 0.3 | | | |
| | | 22–34 | Snow-ice | 6.29 | 8.56 | 0.3 | 6.81 | 102.9 | 0.2 |
| | | 34–42 | Mix-ice | 6.60 | 8.51 | 0.2 | | | |
| | | 42–48 | Cong.ice | 6.56 | 9.92 | 0.1 | | | |
| | April 22 | 0–14 | Snow-ice | 6.25 | 7.53 | 0.3 | | | |
| | | 14-28 | Snow-ice | 6.25 | 7.31 | 0.3 | 6.79 | 92.2 | 0.2 |
| | | 28–38 | Cong.ice | 6.51 | 6.93 | 0.2 | | | |

| Year | Date | depth | ice type | pH | EC | Chl a | water pH | water EC | water Chl a |
|---|---|---|---|---|---|---|---|---|---|
| | April 26 | 0–7.5 | Snow-ice | 6.24 | 7.53 | 0.5 | 6.74 | 84.9 | 0.6 |
| | | 14.5–24.5 | Cong.ice | 6.58 | 7.71 | 0.4 | | | |
| | April 29 | 0–6 | Snow-ice | 6.25 | 7.60 | 0.5 | 6.79 | 74.1 | 1.7 |
| | | 18–22 | Cong.ice | 6.58 | 8.72 | 0.4 | | | |
| | May 3 | 0–2 | Snow.ice | 6.47 | 10.3 | 0.5 | 6.98 | 81.2 | 1.7 |
| 2018 | April 12 | | | | | | 6.86 | 97 | |
| | April 13 | | | | | | 6.87 | 97 | |
| | April 14 | | | 6.39 | 6 | | 6.83 | 97 | |
| | April 15 | | | | | | 6.81 | 97 | |
| | April 18 | | | 6.30 | 6 | | 6.80 | 96 | |
| | April 20 | | | | | | 6.77 | 81 | |

The mean ± standard deviation of pH, EC in the ice were $6.44 \pm 0.28$ and $11.4 \pm 5.77$ S cm$^{-1}$ in the
winter 2021–2022. In lake water, the corresponding quantities were $6.82 \pm 0.09$ and $92.5 \pm 12.7$ S cm$^{-1}$.
The mean value of EC in snow-ice and congelation ice were of the same order of magnitude but by one
order of magnitude lower than that in the lake water, in exact form EC(ice) = 0.12·EC(water). The same
result was found in the winter of 2018–2019. The mean value of pH in snow-ice was 6.34, a little lower
than 6.63 in congelation ice. The deposition of acidic substances from the atmosphere was the
background for the low pH of snow-ice. This can also be confirmed by the data of EC on April 14. EC
of ice decreased with the ice melting, but increased after the snowfall on April 5. The mean value of Chl
$a$ content in ice was less than 0.5 g L$^{-1}$, 0.35 times of that in lake water.
Figure 5 shows pH, EC and Chl $a$ in snow-ice, congelation ice and lake water with the ice melting
process. The mean pH and EC in ice and lake water decreased with ice decay. However, they slightly
increased after the slush layer appeared on April 26. The main reason is that after the slush layer
appeared, some lake water flooded into the slush layer, and the high pH and EC in lake water caused
their slight increase in the ice. EC was lower in congelation ice than in snow-ice at the beginning of ice
decay, and after April 8 they became very close because of melting effects. Chl $a$ was very low since
the ice limited the transmission of light, and photosynthesis in ice and water was very weak. But as the
thickness of the ice decreased, the transmission of light increased, primary production continued to rise
and the content of Chl $a$ in the ice and water increased gradually. Algae can grow in a slush layer within
snow-ice, but not in consolidated ice because of lack of liquid water for living organisms. The present
article reported that the Chl $a$ in snow-ice is greater than in congelation ice but less than in water.





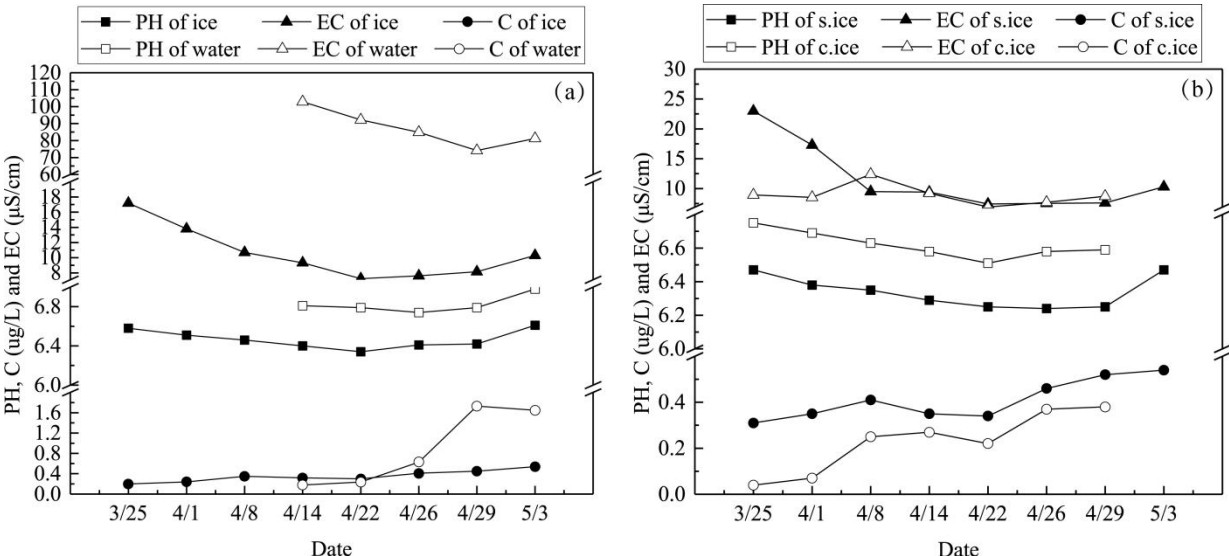


Figure 5. The mean pH, EC and Chl *a* in ice and lake water in 2022 (left) and the mean pH, EC and Chl *a* in snow-ice

(s.ice) and congelation ice (c.ice) (right).
**4 Heat budget**
The heat content of lake ice was used to analyze the observations of ice melting. The heat fluxes include
solar radiation, terrestrial radiation, turbulent air-ice fluxes at the surface, precipitation, and heat flux
from the water body to ice bottom (e.g., Leppäranta, 2015). In the melting period, we consider the
volume of ice per unit area ($V$), expressed by the ice thickness ($h$) and porosity ($v$) as $V = (1 - v)h$. It
is assumed that in the melting stage the ice is isothermal with the temperature at the melting point. The
mass balance is then given by (Leppäranta et al., 2019)
$$\rho_i L_f \frac{dV}{dt} = -(Q_0 + Q_A + Q_w),$$    (3a)
$$\rho_i L_f (1 - v) \frac{dh}{dt} = -(Q_0 + Q_w),$$    (3b)
$$\rho_i L_f h \frac{dv}{dt} = Q_A,$$    (3c)





Where $\rho_i$ is ice density, $L_f$ is latent heat of freezing, $Q_0$ is surface heat balance, $Q_w$ is heat flux from
water, and $Q_A$ is absorption of solar radiation in ice. At $\nu = \nu^* \sim 1/2$, ice breaks due to its own weight
and the remains melt then fast.
In the melting period, the surface heat budget is dominated by the radiation balance with solar radiation
having a key role (Wang et al., 2005: Jakkila et al., 2009). The input fluxes in Eq. (3b) can be estimated
by (see Leppäranta, 2015)
$$Q_0 = k_0'(t) + k_1(T_a - T_0), \qquad (4)$$
where $k_0'$ depends on solar radiation and therefore on time, and $k_1 \sim 15$ W m$^{-2}$ °C$^{-1}$. It is assumed that $k_0'$
takes half of the solar radiation while the other half is let to penetrate the near-surface layer. Then we
obtain a representative, climatological $k_0'$ by interpolation from the mean values of –48 W m$^{-2}$ in March,
$-34$ W m$^{-2}$ in April and 4 W m$^{-2}$ in May based on Leppäranta (2015). The total modelled surface
melting became 25 cm that is rather close to the result (33 cm) obtained from the ice structure analysis
(Table 3). The value of $k_1 \sim 15$ W m$^{-2}$ °C$^{-1}$ corresponds to the degree-day coefficient of 0.43 cm (°C·d)$^{-1}$
$^{1}$, which is close to the usual degree-day coefficient in hydrological forecasting (Leppäranta, 2015).
The question is then internal melting and bottom melting which depend on the solar radiation. We have
(see Leppäranta et al., 2019)
$$Q_A = (1 - \alpha)\gamma\left(1 - e^{-\lambda h}\right)Q_{s0}, \qquad (5)$$
$$Q_w = Q_{w0} + c(1 - \alpha)\gamma e^{-\lambda h}Q_{s0}, \qquad (6)$$
where $\alpha$ is albedo, $\gamma$ represent the fraction of light in solar radiation, and $\lambda$ is the light attenuation
coefficient. The climatological value of solar radiation in April is $Q_{s0} = 150$ W m$^{-2}$. Taking the optical
parameters as $\alpha = 0.5, \gamma = 0.5, \lambda = 1$ m$^{-1}$, as the representative solar flux in April, we have $Q_A = 11$
W m$^{-2}$ corresponding to melt rate of 0.32 cm d$^{-1}$, more than 0.18 cm d$^{-1}$ obtained from the ice structure
data. To evaluate the heat flux from the water, we can take $c = 0.3$ (Leppäranta et al., 2019), and then
$Q_w = Q_{w0} + 9.1$ W m$^{-2}$, and according to estimate of $Q_w = 13$ W m$^{-2}$ in Section 3.1, we have $Q_{w0} =$
3.9 W m$^{-2}$ that may look a bit large but can be explained by the inflow from brooks into the bay.





Thus, the comparison between ice structure and heat balance gives satisfactory agreement in the view of
large uncertainties in both data sets. The heat balance gave the triple (surface melting, internal melting,
bottom melting) as (25 cm, 14 cm, 11 cm), while the observed result was (33 cm, 8 cm, 22 cm). There
was not good boundary layer data for above or below ice, and the optical parameters are only roughly
known. It is concluded that the field data and heat budget were consistent within the limits of accuracy
of observations. This means that the heat budget can be used to assess the melting of the ice and further
predict the breakup of the ice.
In April 2018, ice thickness was 35 cm on the 12th, and ice breakup took place on the 25th. The last
five days are not known for the evolution of the ice cover, but in 12–20 the surface melting was 5.3 cm,
bottom melting was 9.4 cm, and internal melting was 6.9 cm (Table 4). With the melting formula (4–6)
and mean air temperature over $12-20$ April of 5.7 °C, we have the surface melting 11 cm, bottom
melting 5 cm, and internal melting 2 cm. Again, these numbers have large uncertainty due to data
limitations, but it is seen that there is certain consistency between ice structure and heat budget data.
**5 Discussion**
**5.1 Ice season and interannual variations of ice breakup date**
Ice phenology time-series includes ice freezing days and the ice freezing and breakup dates. Climate
change studies based on ice phenology have been conducted for many lakes. Field observed ice data are
very important for many single and multiple variable regression analyses used to develop regression
models and physical models to predict the ice phenology (George, 2007; Williams et al., 2004; Stefen
and Fang 1997). In the ice season 2021–2022, the air temperature fell below the freezing point of water
at mid-November (Fig. 6) and primary ice formed in the study lake at the end of December (Shumskii,
1956). Thereafter congelation ice grew steadily downward, and snow-ice formed on the top mostly due
to flooding of the ice. The seasonal maximum thickness of 55 cm was reached in late March. The ice
freezing days and the date of ice freezing are affected by parameters that determine heat storage and
release of the water body. In contrast, the ice breakup date depends on solar radiation and the
characteristics of the ice and snow. The snow cover strengthened the albedo and blocked the exchange



of heat between the atmosphere and ice that reduced congelation ice growth rate as well as prevented

deterioration of the ice under snow. In the middle of March, the daytime air temperature started to be

above the freezing point and snow melted first and disappeared by the end of March. Then, ice melting

started, paused for a week due to snowfall on April 5 with a thin new snow-ice layer on ice. After mid-

April ice melted by 2–3 cm d$^{-1}$ and finally disappeared on May 5. The entire ice season lasted 149 days

and the decay period was 42 days.



**Figure 6. Air temperature and freezing data on Lake Pääjärvi during the winter of 2021-2022. (a) the max, mean and min daily air temperature from November 1 to May 30, (b) Snowfall in ice decay period, snow and ice thickness measured by Finnish Environment Institute (SYKE) from November 1 to May 6, and ice thickness measured by this research from March 25 to May 5.**



Climate variations have a major impact on ice season characteristics; in other words, ice season
characteristics are sensitive indicators of climate. In the period from 1970 to 2022, the average length of
ice season was 130 days in Lake Pääjärvi, and the standard deviation of 25 days showed a great
dispersion. The ice breakup was on average April 25, with a standard deviation of 12 days. The time
series is short but shows ice breakup becoming earlier in the last 50 years (Fig. 7). However, the
interannual variability of the ice breakup date is quite high. In 1970–1990 the change was about 5 days
per decade that is more than could be explained by global warming and the reason remains unclear. In
general, in southern Finland the trend has been 0.5–1 days earlier breakup per decade. Results on the
lake ice breakup date have shown change of only about 3–4 days per 50 years (Bernhardt et al., 2011;
Magnuson et al, 2000). In an arctic tundra in Finland, Lake Kilpisjärvi, the trend from 1964 to 2008 was
2.2 days over 50 years towards earlier ice breakup (Lei et al., 2012). Reduced ice freezing days and
earlier ice breakup could have a potentially widespread implications on 50 countries (Sharma et al.,
2019). The loss of lake ice could lead to a reduction in the availability of fresh water due to increased
rates of evaporation, as well as ice cultural and socio-economic impacts for lake ice recreation, such as
ice fishing and skating.

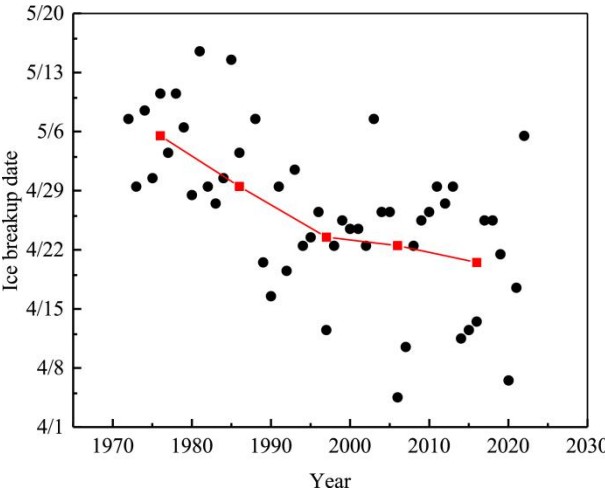


**Figure 7. The ice breakup date of Lake Pääjärvi from 1970 to the present, the black dots indicate the ice breakup**
**date while the res dots indicate the ice breakup date averaged in every 10 years. Data source: Lammi Biological**
**Station.**



## 5.2 Comparisons with ice melting

Ice melting is related to air temperature, solar radiation, albedo, lake bathymetry and morphology, as
well as the ice structure in particular on the fractions of clear congelation ice and opaque snow-ice.
Melting begins after the net radiation becomes positive and takes place at the surface, interior and
bottom depending on the surface heat fluxes and the absorption of solar radiation within the ice. Surface
melting is not only reflected in a reduction in the thickness of ice but also in visible changes of the
surface of the ice cover.
The melting of ice is illustrated by the accumulated melting thickness for the total and the surface,
bottom, and internal portions separately (Fig. 7). In 2022, the surface melting was greater than the
bottom melting, while in 2018, it was the opposite. The main reason was that the ice structure was
different in these two ice years. In 2022, the snow-ice layer accounted for 60 % of the ice cover, while
in 2018, the fraction was only 15 %. However, it can be seen from Fig. 7 that the melting of the surface
layer and the bottom layer were increasing at the same time, and the melting rate was gradually
increasing due to the weather was getting warmer and warmer and solar radiation increased
continuously.

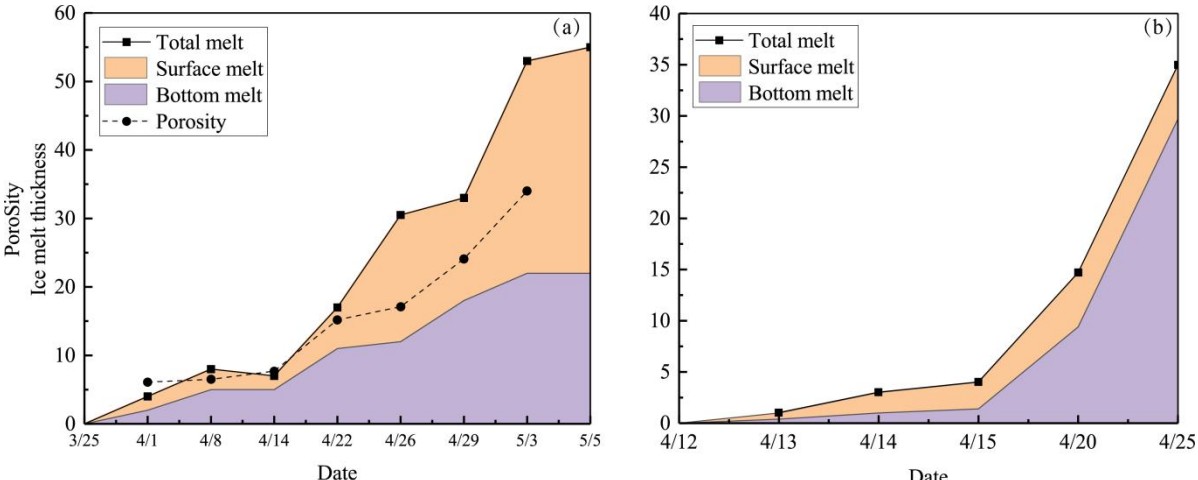

**Figure 8. Accumulated ice melting and porosity in 2022 (left) and 2018 (right). Porosity was not recorded in 2018.**

Overall, the mean ice melting rate was 1.31 cm d$^{-1}$ in 2022. After the new snow had disappeared, the
surface and bottom melt rates were 1.63 cm d$^{-1}$ and 0.8 cm d$^{-1}$, respectively. In Lake Kilpisjärvi, Arctic
tundra, the melting of ice had similar features with Lake Pääjärvi. In 2014 with normal weather



conditions the rate was close to 2022 but larger in the very warm year 2013 (Leppäranta et al. 2019). In
boreal lakes at $61 - 62°$ N, numerical modelling in Lake Vanajavesi (Yang et al. 2012) and field
investigation in Lakes Vendyurskoe (Leppäranta et al. 2010) and Pääjärvi (Leppäranta et al. 2009) gave
similar melting rates as here. The main results on ice melting, if further generalized, will provide the
necessary quantitative information for estimating the seasonal response of ice to climate change.



**Table 6. ice data in other boreal lakes.**

| Lake | Date | Location | Average depth | Maximum depth | Ice season | Ice thickness | Ice melting rate |
|---|---|---|---|---|---|---|---|
| Vendyurskoe<br>Leppäranta et al. (2010) | 2006<br>2007 | 62°10′ N<br>33°10′ E | 5.3 m | 13.4 m | 180–190 days<br>Breakup date 10–20 May | 60-69 cm | 2006:<br>1.2 cm d$^{-1}$ on the surface layer,<br>0.2 cm d$^{-1}$ on the bottom layer;<br>2007:<br>1.2 cm d$^{-1}$ on the surface layer,<br>0.8 cm d$^{-1}$ on the bottom layer. |
| Kilpisjärvi<br>Leppäranta et al. (2019) | 2013<br>2014 | 69°03′ N<br>20°50′ E | 19.5 m | 57 m | 4–6 months<br>Breakup date in June | 77–114 cm | 2013:<br>2.9 cm d$^{-1}$ on the surface surface,<br>1.0 cm d$^{-1}$ in internal,<br>0.5 cm d$^{-1}$ on the bottom layer;<br>2014:<br>0.8 cm d$^{-1}$ on the surface,<br>1.0 cm d$^{-1}$ in internal,<br>0.1 cm d$^{-1}$ on the bottom layer;<br>Mean melt rate: 1.3cm d$^{-1}$. |
| Pääjärvi<br>Wang et al. (2005);<br>Jakkila et al. (2009) | 2004<br>2006 | 61°04′ N<br>25°08′ E | 14.8 m | 87 m | 4–6 months | 30–80 cm | 1.25 cm d$^{-1}$ on the surface layer. |
| Vanajavesi<br>Yang et al. (2012) | 2008 | 63°13′ N,<br>24°27′ E | 7 m | 24 m | 4–6 months | 45–60 cm | Mean melt rate: 1.3 cm d$^{-1}$ |

Ice thickness and temperature are the simulated ice properties in lake ice physical models (Ashton, 1986;
Shirasawa et al., 2006; Leppäranta, 2009). This works during the ice growing season, but during the
melting season, the variation of ice thickness does not tell of internal melting, for which porosity data
are needed. Internal melting changes the structure of the ice, and once the porosity reaches around 50 %,
the ice cannot bear its own weight, breaks, and disappears rapidly (Leppäranta et al. 2019). A study in



Lake Pääjärvi in 2004–2006 found that the breakage resulted at the porosity of 45 % (Leppäranta et al.
2009). The present work measured ice density in the 2022 melting period, and the porosity
corresponding to the measured ice density was used as the porosity estimator. The porosity increased
with the ice melting (Fig. 7). The density of pure ice is 917 kg m$^{-3}$ and the estimated porosity was 34 %
on May 3, and the ice broke up on May 5 when the porosity of the ice could have been $40-50$ %
consistent with Leppäranta et al. (2019). The internal deterioation is also a possible reason of the error
about the ice rupture model. Yang et al. (2012) modelled ice breakup date turned out to be 12 d too late.
The internal deterioration of the ice cover becomes extremely important, not only for the physics of ice,
but also for spring ecology and the practical issues related to ice strength.
In spring, internal melting of ice can cause a significant reduction of the ice strength. This has two
important consequences. First, the bearing capacity of ice decreases. The bearing capacity scales as
$\sigma_f h^2$, where $\sigma_f$ is the flexural strength. During the melting period, ice thickness decreases due to
surface and bottom melting while ice strength decreases from internal melting. Due to the positive
albedo feedback in the melting, the ice cover becomes patchy for its strength and the bearing capacity is
largely unpredictable, that is a severe safety issue. Secondly, resistance of lake ice cover to breakage
scales with $\sigma_c h/L$, where $\sigma_c$ is compressive strength and $L$ is the length scale of lake size. Decreasing
thickness and strength may lead to breakage and ice movement on shores, where damage can be caused
since the strength still is finite.
The deterioration of ice cover is not necessarily accompanied by an overall thinning of the ice cover.
Since most engineering guidelines for bearing capacity are based on ice thickness and strength in
relation to complete structural integrity, it is important to understand under what conditions these
guidelines may be misleading. Therefore, it is necessary to know the ice porosity due to affections the
level of force exerted on the structures. There are several models to relate porosity to failure stress
(Ashton, 2012; Bulatov, 1970). Since boundary conditions of the crystals, and the density and porosity
of ice need to be used in the model, the present study is of great help to the development of this kind of
models.
The melting at both ice boundary and in ice interior was investigated in this study based on the field
observation and calculation of heat budget. The results on the heat budget during the ice melt period can



reveal the physical mechanisms behind seasonal formation and deteriorate of ice cover in different
climatic conditions. The heat and mass transfer at the ice-water interface is the least studied among
these mechanisms.The greater the heat flux from the water, the smaller ice thickness and the earlier the
breakup time. For example, a heat flux of 1 W m$^{-2}$ melts about 1 cm of ice every month, and the
average melting rate of 1.3 cm d$^{-1}$ ice breaks up about 1d earlier. The present study gave the heat flux
corresponding to the bottom melting in two ice seasons, and the more transparent ice that allowed more
sunlight penetration through ice in 2018 obtained larger heat flux than 2022. Compared with Lake
Kilpisjärvi (Leppäranta et al., 2019), our fluxes were less than in a very warm year (15–20 W m$^{-2}$) but
more than in a normal year (5–10 W m$^{-2}$). Much of the earlier literature has reported of smaller values
at later stages of ice melting. Bengtsson et al. (1996) obtained for a number of small Swedish lakes the
heat flux from water to ice ranging within 5–7 W m$^{-2}$ in March–April. However, Jakkila et al. (2009)
reported the heat flux values in Lake Pääjärvi as 12 W m$^{-2}$ during the final stage of ice melting that is
very close to the present results. Leppäranta et al. (2010) reported the heat flux of 7–29 W m$^{-2}$ in late
spring in the boreal Lake Vendyurskoye. The lake size may be the reason for the differences in water-
ice heat fluxes, since in general the heat content is smaller and circulation weaker in small lakes.
However, bottom melting remains the most uncertain component of the heat budget, and more field data
and future research are needed particularly on the influence of the stage of ice melting, state of the
under-ice boundary layer, and the amount of heat stored in the water during winter (Kirillin et al., 2018).

## 5.3 Ice melting impact on geochemistry

Deterioration of lake ice takes place at the top and bottom boundaries and in the interior. Porous melting
ice is permeable to water, so that meltwater can flow down from top and lake water may penetrate to
pores from below. These processes also influence the stratification of the surface water layer under the
ice. The significance of meltwater to underwater chemistry and biology has not been much studied in
lakes, apart from the density-driven stratification effect (Kirillin et al., 2012). Mathematical models for
deterioration exist (Leppäranta, 2015) but are not in wide use, maybe because the melting period is
short and once begun it progresses more or less steadily. Especially the gechemistry properties during
ice melting period are rarely reported.





Lake Pääjärvi was studied for ice and water geochemistry in mid-winter in 1996–1998 (Leppäranta et
al., 2003). They measured the mean values of EC in snow, ice, and water as 16.5, 13 and 108 S cm$^{-1}$
with the ranges of 4–28, 9.5–28 and 79–208 S cm$^{-1}$, respectively, and pH was 6.7 for ice and 6.6 for
water. The value of pH in snow was typically 0.2 pH units lower than in ice. In this study, in 2022 the
mean EC was in the snow-ice, congelation ice and lake water 13.0, 9.47 and 93.1 S cm$^{-1}$ with the ranges
of 7.53–31.1, 6.93–16.4 and 74.2–102.9 S cm$^{-1}$, respectively. It is worthy to notice that both pH and EC
in ice melting period decreased with the ice decay and were smaller than in the mid-winter. After the
melting started, the light under the ice increases due to the changes in the ice structure. The increase
photosynthesis enhances $CO_2$ consumption and the pH of the water should be at a relatively higher level.
There are two reasons for the lower pH: first, meltwater in ice was injected into water; second,
biological activities under the ice became active with the rising of water temperature, and there was a
surge of phytoplankton under ice resulting in an increase of $CO_2$, which leads to a continuous decline in
pH value. But after the slush layer appeared, both pH and EC in ice increased due to lake water flushing.
Based on the data of the inflows from brooks into the study bay. The current was almost static by April
21, whereas the inflow corresponded to 17 % of the water volume of the bay April 21–25. This means
the geochemistry of the lake water was also affected by the brooks. Figure 9 showed the pH and EC of
the inflow brooks and the results revealed the consistent changes with the lake.

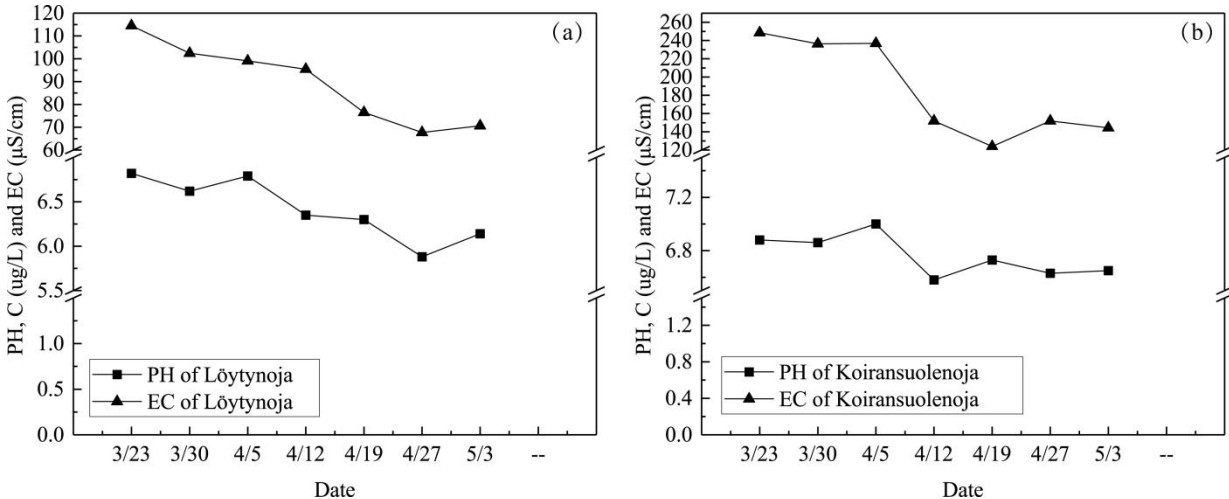


**Figure 9. The pH and EC in Löytynoja (left) and Koiransuolenoja (right).**





The mean EC in ice was of the same order of magnitude but one order of magnitude lower than the lake
water EC in both studies. The pH in snow-ice and congelation ice is a little lower than that in lake water
in 2022 (Fig. 5). Flooding of lake water on ice and atmospheric deposition mostly imported the
impurities into the ice cover. Snow-ice was formed of the snowfall with the melt-freeze cycles, flooding
of the ice or liquid precipitation. Therefore, the deposition of acidic substances in the atmosphere was
an important reason for the lower pH of snow-ice. The same result was found in 2018. The chl $a$ is an
indicator of phytoplankton biomass which can directly and quickly reveal the enrichment status of
phytoplankton (Gradinger, 2002; Tedesco et al., 2012). Chl $a$ was less than 0.5 g L$^{-1}$ in ice and was
lower than in the lake water. During the last two weeks of ice decay, water Chl $a$ varied between 0.2
and 1.7 g L$^{-1}$ which is of the same order of magnitude reported by previous research (Leppäranta et al.,
2003; Vehmaa et al., 2009). The mean Chl $a$ in ice was less than 0.5 g L$^{-1}$, 0.35 times of the lake water
Chl $a$. Leppäranta et al. (2003) also reported the ratio of the Chl $a$ in ice and water was 0.16. pH, EC
and chl $a$ are important indicators of water environmental quality. These environmental factors are not
only the physical parameters of water environment, but also affect the physiological state of aquatic
organisms, which will guide and predict the changes of biological structure in the water during the
melting season.

## 551   6 Conclusions

The formation and decay of ice cover are changing under the influence of global warming. Due to the
increasing attention to the climate impact on mid and high latitude lakes, more and more studies have
been conducted on lake ice. Since it is very difficult to do fieldwork during the melting period, there are
only few field data over the full ice decay period. The present has filled to this gap of knowledge
focusing on the ice decay in Lake Pääjärvi, a boreal lake in southern Finland, in 2018 and 2022.
Lake ice melting and breakup form a fast, nonlinear process. The process is difficult to study in the field
due to safety issues, and therefore relatively little is known about its details. The field observations were
made in Lake Pääjärvi during the ice decay periods in 2018 and 2022. Ice monitoring was based on foot,
hydrocopter, and boat, and a full time-series was obtained of the evolution of ice thickness, porosity,
structure and geochemical properties through the melting period.



The results show how melting of lake ice takes place at the surface and bottom and in the interior
simultaneously, and as a result ice thickness decreases and ice porosity increases. This drastically
changes the physical properties of ice with consequences to the physics, chemistry, and biology of the
waster body. The mechanical strength of ice decreases that has consequences to the bearing capacity of
ice and ice forces. Also, weakened lake ice may be broken and pushed onshore by winds that causes
shore area erosion and forces on man-made structures such as piers and navigation marks. The results
are important for further development of numerical models towards more realistic physical presentation
of the ice thickness and porosity during the decay period. This is well supported by the consistency
between the field data of ice structure and thickness and the heat budget.

*Data availability.* The routine meteorological and hydrological data are available at:
https://www.syke.fi and https://www.fmi.fi. The ice samples data applied in this work can be accessed
by the link: https://doi.org/10.5281/zenodo.7342770.
*Author contributions.* YZ conceived the study with ML and wrote the paper. YZ performed the field
work and lab work with contributions from LM, MF, JL, SS, and JA. All co-authors discussed the
results and edited the manuscript.
*Acknowledgements.* We are grateful to the Lammi Biological Station and Institute of Atmospheric and
Earth Sciences (University of Helsinki) for providing help with sample collection and processing.
Thanks to Esa-Pekka Tuominen, Joni Uusitalo and Riitta IIola for helping the fieldwork and laboratory
work. Thanks to Lauri Arvola for helping to edit the manuscript. This work was financially supported
by the National Key Research and Development Program of China (Grant No. 2019YFE0197600), the
National Natural Science Foundation of China (Grant No. 52211530038) and the Academy of Finland
(350576), Personal grant to Yaodan Zhang by China Scholarship Council (CSC).
*Competing interests.* The authors declare that they have no conflict of interest.





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
