# Peer review of "A field study on ice melting and breakup in a boreal lake, Pääjärvi,"

_The Cryosphere, 2022_

## Referee Comment (RC2)

The manuscript focused on the melting regime of lake ice, which has been seldomly investigated despite its crucial impacts on ice break-up, air-lake mass and heat exchange, and lake habitats and ecosystems. The manuscript, using varied technologies, investigated the melting rate, the evolutions of lake ice texture, pore, and crystalline structure, and the concurrent pH, EC, and Chl-a profiles with their relationships with the decaying ice. These observational results are quite important to understand the ice breakup regime and its effects on lake habitats and ecology.

However, the manuscript is too long and poorly-structured and is suffered to linguistic issues, so it is not easy to follow what the authors wanted to deliver to readers. E.g., there are a number of repetitions of sentences in the text, which can be removed or shortened. I am not sure of if the section 5.1 should be presented in *Results* or *Discussion*. I recommend that the manuscript should reorganized and carefully language-checked when revising.

Intuitively, the manuscript is more like a report presenting the techniques and straight results than a scientific paper. It should be more focused and tightly compact on what you found and why. In short, the authors observed the melting rate, internal texture with estimated porosity, and basic ice+water geochemistry. It would better if the manuscript directly present the methods and results and discuss what controls the surface, bottom and interior melting, how these melting are related to the evolution of pH\EC\chl-a, and how it expands/deepens our knowledge on lake ice decay and accompanying changes in lake environment, so the interannual variations of ice breakup date and bearing capacity discussion seem redundant and not closely related to the topics.

Below, I have some specific comments and suggestions, which can be considered by the authors during revision.

[1] L31: by/through altering the heat, mass…

[2] L36: the under-ice living conditions or the living conditions under the ice

[3] L43: please specify what the two major practical problems are.

[4] L53-54: I guess "by about one week over 100 years" is actually true for e.g. boreal lakes? Should be specific here.

[5] L67: the primary production

[6] L69: limits the proper assessment of the impacts of…

[7] L71-90: I suggest this paragraph can be divided to several short paragraphs according to their key points. At its current status, it is not easy to clearly understand what the authors want to tell to the readers.

[8] L91-98: I guess in this paragraph the authors should introduce briefly the scientific issues to be targeted, what work and analysis has been done, and what problems could be resolved. The snow and ice conditions may be better to be presented in the "2.1 study site".

[9] L125: delete "lateral"?

[10] L164: delete "*Measurements of ice density can be found in several studies (Timco and Frederking, 1996).*"

[11] L169-176: please add the measuring accuracy for each variable.

[12] Table 1: what did x and z denote?

[13] Figure 2: please add a scale to these maps.

[14] L221: due to

[15] Eq. (1): there must be mistakes in this equation since it was not consistent with the description bellow it and I cannot see $L_f$ and $\Delta t$ in the formula.

[16] L295-296: delete "*In practice it is difficult to determine the freeboard/draft ratio as it requires an order of one-millimetre accuracy for the freeboard.*"

[17] Table 5: It would be better if these data can be presented in a plot/plots (i.e., vertical profiles), which can show clearly the vertical structures and temporal variations of EC, pH, and Chl-a.

[18] L322: how did the data of EC on April 14 confirm the deposition of acidic substances from the atmosphere? And why did EC of ice increase after a snowfall?

[19] L333-334: "Algae can grow in a slush layer within snow-ice, but not in consolidated ice because of lack of liquid water for living organisms." However, Fig. 5b shows clearly the Chl-a content in congelation ice increases gradually as melting proceeds. The increase in Chl-a content within the ice is likely to result from the increasing solar radiation and/or the decreasing surface albedo rather than the thinning ice cover.

[20] Eqs. (3): it would be better if you present briefly the physical meaning of each formula.

[21] L359: 4 W m$^{-2}$ can not be a half of the incoming solar radiation in May.

[22] L363: does the bottom melting depend on the solar radiation?

[23] L367: what do you mean by "γ represents the fraction of light in solar radiation"? I guess it is the fraction that penetrates through the ice surface.

[24] L368: by Qs0=150W m$^{-2}$, is it the daily-averaged value? It looks like a daytime-averaged value in April. If this is true, Qs0 ≈ 75 W m$^{-2}$, the melt rate can be 0.16 cm d$^{-1}$, close the observed rate of 0.18 cm d$^{-1}$.

[25] Eq.(6): Could please give a brief physical background of this equation? What do the terms mean at the right-hand side?

[26] L389: the second "freezing" should be "melting"

[27] L397: what do you mean by "the ice freezing days"? freezing duration, or ice-covered duration?

[28] L433: after the net radiation becomes positive? The net radiation is always positive, I guess. And whether or not the surface, interior, and bottom melting take place depends on different conditions of heat balance.

[29] L467: "Yang et al. (2012) modelled…too late", what do you mean?

[30] L482: "Therefore, … on the structures.", what do you mean?

[31] L527: a surge of phytoplankton under ice may indicate a positive net production, which uses CO2 to produce oxygen and biomass, so why it results in an increase of CO2? Could you explain on it a bit more? Maybe the inflow dominates the chemistry regime of the surface layer as is shown in Fig.9?

[32] Conclusions: Usually, in the conclusion section, notable technologies, results and findings should be presented as well as brief implications if any rather than research background and motivation. Key points that were found in present work were missed here. So I recommend to reorganize this section.

[33] L555: The present paper/investigation has filled…

---

## Author Comment (AC1)

Dear reviewer,
Thank you very much for your letter regarding our manuscript entitled "A field study on ice melting and breakup in a boreal lake, Pääjärvi, in Finland". We are truly grateful to the comments and suggestions from you. The manuscript has been carefully revised, and the language has been checked. The point-by-point answers to the comments and suggestions were listed as below.

**Response to Reviewer 1# Comments**

The study presents detailed observations on variations of lake ice properties during two melting periods in a boreal lake. The observations encompass ice structure, porosity, density, biogeochemical characteristics (pH, electrical conductivity, Chlorophyll a) and were performed at daily to weekly intervals covering the latest stages of the ice-covered period. The study design and the approach are relevant to the state-of-the-art of the lake ice studies. The newly collected data have a potential to make a valuable contribution to the current knowledge on the mechanisms of the seasonal ice cover melt.

The presentation of the results is however extremely hard to follow, too lengthy and poorly structured. Description of methods pops up in the middle of results presentation, while new introductory information and collateral results, which are only loosely connected to the subject of the study, unexpectedly start the discussion part. The authors should analyze, synthesize, summarize and---finally---present scientific results in a concise, well structured way.

Endless recitation of dry unbound numbers throughout long paragraphs, repeating the information presented in tables, without an integrated analysis of the field information is redundant and superficial. Last but not least: the poor language and style make the study hard to read. Overall, the style reminds that of a plain field report rather than a research paper. Many sentences are barely understandable because of poor English use. Reworking of the text with significant shortening, restructuring and language improvement is strongly recommended. The length of the ms should conform to the amount of the reported results, which suggest shortening it by 1/3-1/2. Below are some specific comments aimed to provide guidance on the revision. The comments are aligned with the text flow. The remarks on language and style are not exhaustive and serve just as the most evident examples.

Response: Thank you very much for your constructive comments and recommendation of the revision. We adjusted the structure of the manuscript to make it easier for readers to understand. We have adjusted our results, discussion and conclusions chapters. In chapter 3 (Results), we added sub-title for section 3.1 (Ice structure) to make it easier to follow. Considering also the comments of Reviewer 2#, in section 3.3 (Ice geochemistry), we presented the data of Table 5 in a vertical plot/plots which can show clearly the vertical structures and temporal variations of EC, pH, and Chl *a*. And we gave a short and clear explanation in the text instead of repeat the dry numbers throughout long paragraphs. We transferred chapter 4 (Heat budget)

to a new section 3.4. We removed all the results from the Discussion chapter to chapter 3 (Result), and give a revised discussion. Finally, based on the manuscript, we have rewritten the conclusions with a clear summary about what we have done, the result and the meaning. We cut the repetitive descriptions in the manuscript and shortened the manuscript by 1/3. And we have checked the language. Please, see our responses to your comments as follows for more details.

Comment:L49 "structure" -> "structures"
Response: We have revised "structure" by "structures".

Comment:L56: "physics of climate sensitivity…" poor wording
Response: We have revised the sentence as: The timing of ice breakup is a question of atmospheric warming and falling albedo (Leppäranta, 2014), and its proper solution requires a quantification of the physical mechanisms that control the melting of ice.

Comment:L61: "seasonal" -> "seasonally"
Response: We have revised "seasonal" by "seasonally".

Comment:L67: "productive" -> "production"
Response: We have revised "productive" by "production".

Comment:L69: remove "the extent to"
Response: We have removed "the extend to".

Comment: L72: "protects the ice." from what? Remove "by its presence"
Response: Removed as suggested.

Comment: L73: "immediately when" -> "immediately after"
Response: Changed as suggested.

Comment: L79: "difficult conditions": difficult for what?
Response: You are right, we didn't make it clear here and we revised it as " Due to the difficult fieldwork conditions on deteriorating ice cover, there has not been much in situ research during the ice decay period. "

Comment:L80: "melt rate" -> "rate of melt" or "melting rate"
Response: We have revised "melt rate" by "melting rate".

Comment:L86: "has reached" -> "reaches"
Response: We have revised "has reached" by "reaches".

Comment:L171: "unifiltered" -> "unfiltered" (?)
Response: Changed as suggested.

Comment: L173: "high accuracy" -> provide the accuracy values.

Response: We have provided the values, for pH (0.01) and EC (0.01 μS cm$^{-1}$).

Comment: L176: "long wavelength" -> provide the wavelength range

Response: We have provided the wavelength (665 and 750 nm) for measuring Chl *a*.

Comment: L192: add the spatial scales and explain spatial scales in Fig. 2

Response: Thanks for your suggestion. We have added the spatial scales and the color scale in Fig.2.

Comment: L195-196: remove "As we can see from Fig 3a-f", add "(Fig 3a-f)" at the end of the sentence.

Response: Considering also other comments of you, we have revised Fig 3a-f. Then we have removed "As we can see from Fig 3a-f", and added "(Fig 3)" at the end of the sentence.

Comment: L200: "became more and more" -> "increased"

Response: We have revised "became more and more" by "increased".

Comment: L207-208: "rachis-shaped" revise wording

Response: We have revised "rachis-shaped" by "cylindrical and spherical shaped".

Comment: L212: Remove "Then"

Response: We have removed "Then".

Comment:L213: "temperature rose…" Temperature of what? The same for L217

Response: We have revised "temperature rose…" by "air temperature increased…".

Comment: L226-236: Fig 3 is not comprehensible and should be shortened to present the essential information only.

Response: Here we wanted to show how the ice structure changed during the melting period. We have adjusted Fig.3, preserving the essential information. Now a typical ice structure, April 1, at the beginning of the melting period is presented in Fig. 3, and the vertical ice structure of ice samples on April 26, April 29, and May 3, which can reflect the changes in ice structure, in Fig. 4.

Comment: L244-254: The paragraph mostly repeats information from Table 2 and should be shortened to 1-2 sentences.

Response: Thanks for your advice. There was too much repetitive information from Table 2 and we have adjusted this paragraph into 3 sentences: "In 2018, the ice decay period began at the end of March, and the final breakup took place on April 25. The thickness of ice was 42 cm on March 30, and on April 12 it was 35 cm with 5.3 cm snow-ice and 29.7 cm congelation ice (Table 2). Snow-ice melted in less than eight

days, and congelation ice melted fast after April 15. On April 24, rain greatly accelerated the melting.

Comment: L262: Equation 1 is wrong.
Response: You are right. We have modified the Equation 1 as:

$$Q = \frac{\rho_i L_f \Delta h}{\Delta t} = 13 \text{ W m}^{-2}, \tag{1}$$

Comment: L264-265: revise the style of the sentence
Response: We have revised the sentence as: The mean internal melt rate was 0.18 cm d$^{-1}$ equivalent ice thickness that was smaller than the surface and bottom melting, attributed to the low light transmittance of snow-ice.

Comment: L277-279: awkward phrasing. Revise the sentence
Response: We have revised the sentence as: The density profiles shifted toward lower level with time while the density always increased with depth (Fig. 5).

Comment: L303 and elsewhere: "S cm^-1" Is it "uS cm^-1"? Check the units across the entire text.
Response: You are right. We actually wanted to write "µS cm$^{-1}$". We have now written "µS cm$^{-1}$" for all conductivity values and we checked all other units across the entire text.

L307: remove double "in"
Response: We have removed double "in".

Comment: L315: shorten Table 5, remove information repeating Fig. 5
Response: Table 5 showed pH, EC and Chl *a* in vertical profiles of ice and in the lake water, and Fig.5 showed the mean value n ice and lake water in 2022. Considering also the comments of Reviewer 2#, we present now the Table 5 data in vertical plots which show clearly the vertical structures and temporal variations of EC, pH, and Chl *a*. Also, we made a short and clear explanation in the text for Fig.6 as follows: The vertical profiles of EC, pH and Chl *a* show that EC was larger near the snow-ice surface than in congelation ice in the early melting stage, but the difference was no more obvious after April 14. pH was always smaller in snow-ice than in congelation ice. Chl *a* content was less than 0.6 µg L$^{-1}$ with the maximum at the snow-ice – congelation ice interface.

Comment: L339, Section 4: The section represents the rare attempt to analyze the observed data beyond their straightforward listing. However, the heat budget model, as presented here, is rather crude and lacks support by background physics of the heat exchange between air and ice surface. The monthly climatic means of solar radiation are too rough for such a model and can be replaced by data from reanalysis or nearby weather monitoring for the actual dates. Assumption of constant albedo is also weakly

supported for the melting periods, especially with snowfall and rain events. Several approaches exist for albedo parameterization, with the simplest ones based on air temperature. Still, one could expect even more sophisticated albedo parameterizations, taking into account the detailed information on ice properties and drone images of the surface conditions. There is no clear model described for the long-wave radiation budget in the ice-air system and for the sensible/latent fluxes at the ice surface. By this, the inconsistency of the oversimplified model with the data (L376-378) is not surprising. The approach should be deeply revised based on the current knowledge on the surface heat budget.

Response: Thanks for your suggestion. We have revised this section based on your comments. The meteorological data from FMI from 2022 was used to analyze the heat budget of ice melting. Solar radiation is now from the daily observed values instead of using the monthly climatic means of solar radiation. The albedo was parameterized as $\alpha = 0.7$ for snow, 0.5 for fry ice and 0.3 for wet ice. After revising the input data for the heat budget model, the results of the model simulation are in good agreement with the observed results. Considering also the comments of Reviewer 2#, we present briefly the physical background physical and meaning of the model and formulae.

Comment: L388, Section 5.1: This section appears absolutely unexpected and does not fit in line with the general flow of the study. Section is an odd mix of newly presented data lacking a thorough analysis of their relevance, statistical significance and processing methods (Figs. 6-7) with unnecessary common places (L411-412), and information on ice phenology from other lakes irrelevant to the subject of the study. The whole section reads inorganic and should be either removed or deeply revised with proper redistribution between methods, results and discussion, including adequate analysis of data reliability.

Response: Thanks for your suggestion. We adjusted the structure of the manuscript and discussion. Section 5.1 has been compressed just to give background to the variability of ice breakup. The date of ice breakup is largely affected by the solar radiation and the thickness and structure ice and snow layers, and the time series shows that ice breakup has became earlier in the last 50 years (Fig.10). A brief comparison with other lakes is also given.

Comment: L430, Section 5.2: "comparisons with ice melting" - comparisons of what? The section title is senseless.

Response: Thanks for your advice. We have revised "comparisons with ice melting" by "Comparisons of ice melting with other lakes"

Comment: L453: "if further generalized…" This generalization _is_ actually what the reader expects from the authors at this point: generalize your results and put them into the context of the present knowledge on the subject. The whole section should be deeply revised. Move Fig. 8 with the accompanying data to Results and write a new, more focused discussion.

Response: Thanks for your advice. We completely revised the discussion, generalize our results and compared the results of this manuscript with other studies carefully. Also, we have moved Fig. 8 with the accompanying data to Results (3.4. Heat budget).

Comment: L526-527: "...a surge of phytoplankton…" - Do you mean a phytoplankton bloom? If yes, can you demonstrate a correlation between pH variations and Cl_a content? This passage reads too speculative and unsupported. Revise it, demonstrating support by data analysis, or remove completely.

Response: We mean a phytoplankton bloom, but it's not possible to give the correlation between pH variations and Chl $a$ content based the present data. We understand that these sentences are somewhat confusing and unclear. Finally, we removed these sentences.

Comment: L555-556: Awkward sentence, should be revised or removed.

Response: We have revised "The present has filled to this gap of knowledge focusing on the ice decay in Lake Pääjärvi, a boreal lake in southern Finland, in 2018 and 2022." by "The field observations were made in Lake Pääjärvi, southern Finland, in 2018 (pilot study) and 2022 (main experiment)." And we have rewritten the conclusions and gave a clear summary about what we have done, the result and the meaning.

Finally, according to the adjustment of the structure and content of this manuscript, the references of this manuscript are also adjusted accordingly.

---

## Author Comment (AC2)

Dear reviewer,
Thank you very much for your letter regarding our manuscript entitled "A field study on ice melting and breakup in a boreal lake, Pääjärvi, in Finland". We are truly grateful to the comments and suggestions from you. The manuscript has been carefully revised, and the language has been checked. The point-by-point answers to the comments and suggestions were listed as below.

**Response to Reviewer 2# Comments**

The manuscript focused on the melting regime of lake ice, which has been seldomly investigated despite its crucial impacts on ice break-up, air-lake mass and heat exchange, and lake habitats and ecosystems. The manuscript, using varied technologies, investigated the melting rate, the evolutions of lake ice texture, pore, and crystalline structure, and the concurrent pH, EC, and Chl-a profiles with their relationships with the decaying ice. These observational results are quite important to understand the ice breakup regime and its effects on lake habitats and ecology.

However, the manuscript is too long and poorly-structured and is suffered to linguistic issues, so it is not easy to follow what the authors wanted to deliver to readers. E.g., there are a number of repetitions of sentences in the text, which can be removed or shortened. I am not sure of if the section 5.1 should be presented in Results or Discussion. I recommend that the manuscript should reorganized and carefully language checked when revising.

Intuitively, the manuscript is more like a report presenting the techniques and straight results than a scientific paper. It should be more focused and tightly compact on what you found and why. In short, the authors observed the melting rate, internal texture with estimated porosity, and basic ice+water geochemistry. It would better if the manuscript directly present the methods and results and discuss what controls the surface, bottom and interior melting, how these melting are related to the evolution of pH\EC\chl-a, and how it expands/deepens our knowledge on lake ice decay and accompanying changes in lake environment, so the interannual variations of ice breakup date and bearing capacity discussion seem redundant and not closely related to the topics.

Response: Thank you very much for your constructive comments and recommendation of the revision. We adjusted the structure of the manuscript to make it easier for readers to understand. We have adjusted our results, discussion and conclusions chapters. In chapter 3 (Results), we added sub-title for section 3.1 (Ice structure), and in section 3.3 (Ice geochemistry) we present the data of Table 5 in vertical plots which show clearly the vertical structures and temporal variations of EC, pH, and Chl *a*. We give a short and clear explanation in the text instead of repeating the numbers throughout long paragraphs. We transferred chapter 4 (Heat budget) to a new section 3.4. We removed all the results from chapter 4 (Discussion) to chapter 3 (Result) and give a revised discussion. Finally, we have rewritten the conclusions and give a clear summary about what we have done, the result and the meaning. We cut

the repetitive descriptions in the manuscript and shortened the manuscript by 1/3. And we have checked the language. Please, see our responses to your comments as follows in more details.

L31: by/through altering the heat, mass⋯
Response: We have revised "Lake ice affects the local weather altering the heat, mass⋯" by "Lake ice affects the local weather by altering the heat, mass⋯".

L36: the under-ice living conditions or the living conditions under the ice
Response: We have revised "the living conditions under-ice" by "under-ice living conditions".

L43: please specify what the two major practical problems are.
Response: Thanks for your suggestion. In fact, we have given the two major practical problems in the manuscript, but we didn't make them clear. We revised this part as: "There are two major practical problems with melting lake ice due to the loss of strength caused by the ice deterioration (Ashton, 1985; Leppäranta, 2015; Masterson, 2009). First, ⋯⋯. Second, ⋯⋯". Then it is clear for readers.

Comment: L53-54: I guess "by about one week over 100 years" is actually true for e.g. boreal lakes? Should be specific here.
Response: Yes, you are right. Here we actually wanted to say in boreal lakes. We have revised "by about one week over 100 years" by "by about one week over 100 years in boreal lakes".

L67: the primary production
Response: We have revised "productive" by "production".

L69: limits the proper assessment of the impacts of⋯
Response: We have revised "properly" by "proper".

Comment: L71-90: I suggest this paragraph can be divided to several short paragraphs according to their key points. At its current status, it is not easy to clearly understand what the authors want to tell to the readers.
Response: Thanks for your advice. L71-90 was somehow confusing to read, we divided it into 2 paragraphs according to their key points. The first paragraph explained how the melting of ice occurs. The second paragraph showed how the internal melting and bottom melting occurs and the influence of internal melting and bottom melting. (Which can be seen in L70-87.)
  Due to the difficult fieldwork conditions on deteriorating ice cover, there has not been much in situ research during the ice decay period. A snow cover delays the melting by its high albedo and low transmissivity of light (Ashton, 1986; Leppäranta, 2015; Warren, 1982). When the ice cover is snow-free, sunlight penetrates to the ice and through the ice. The ice warms up and melts inside, the under-ice water is heated,

and the surface heat balance determines whether surface melting takes place (Kirillin et al., 2012). Ice impurities are released from melting ice into the water that changes the water environment. The under-ice light is also used for primary production, which normally peaks after ice breakup.

The present knowledge of the melting rate of ice is limited to a few studies, showing typical values of $1-3$ cm d$^{-1}$ in terms of equivalent ice thickness. Melting takes place at the top and bottom boundaries and in the interior depending on the weather conditions (Jakkila et al., 2009; Leppäranta et al., 2010, 2019; Wang et al., 2005). It has been found that the light transmittance changes with internal melting that has influence on further melting. Internal melting also opens channels for flushing the ice by surface meltwater and lake water. When the porosity of ice reaches the level of around 0.5, the ice cover collapses by its own weight and disappears rapidly (Leppäranta et al., 2010, 2019). Bottom melting is caused by the heat flux from water that can be large in spring due to the solar heating of the under-ice water (Jakkila et al., 2009; Shirasawa et al., 2006).

Comment: L91-98: I guess in this paragraph the authors should introduce briefly the scientific issues to be targeted, what work and analysis has been done, and what problems could be resolved. The snow and ice conditions may be better to be presented in the "2.1study site".

Response: Thanks for your suggestion. We have rewritten this paragraph as follows:

We examine here the decay of ice in a boreal lake, Lake Pääjärvi, in southern Finland by field surveys in two years, 2018 and 2022. The objective was to analyse the ice melting process for the evolution of ice thickness and porosity as well as for the changes in ice and water geochemistry. The structure and properties of ice experienced remarkable changes during the decay process, and significant melting occurred in the surface and bottom and in the interior. Flushing of ice by meltwater and lake water caused changes to ice and water geochemistry. A deeper knowledge of the ice decay is needed for modelling the lake ice decay, particularly for ice engineering issues, and for understanding the physical and geochemical conditions for ecology of freezing lakes in spring. This paper gives the final results of the Lake Pääjärvi field program.

Comment: L125: delete "lateral"?
Response: Yes, we have delete "lateral".

Comment: L164: delete "Measurements of ice density can be found in several studies (Timco and Frederking, 1996)."
Response: We have delete "Measurements of ice density can be found in several studies (Timco and Frederking, 1996)."

Comment: L169-176: please add the measuring accuracy for each variable.
Response: Thanks for your advice. We have provided the accuracy values for pH (0.01) and EC (0.01 μS cm$^{-1}$).

Comment: Table 1: what did x and z denote?

Response: We have z no more and explain that "x stands for no data."

Comment: Figure 2: please add a scale to these maps.

Response: Thanks for your suggestion. We have added the spatial scales in Fig. 2. And we also added the color scale in Fig. 2.

Comment: L221: due to

Response: We have revised "due" by "due to".

Comment: (1): there must be mistakes in this equation since it was not consistent with the description bellow it and I cannot see $L_f$ and $\Delta t$ in the formula.

Response: Yes, we have revised the formula (1) in to " $Q = \frac{\rho_i L_f \Delta h}{\Delta t} = 13 \text{ W m}^{-2}$ ".

Comment: L295-296: delete "In practice it is difficult to determine the freeboard/draft ratio as it requires an order of one-millimetre accuracy for the freeboard."

Response: We have deleted "In practice it is difficult to determine the freeboard/draft ratio as it requires an order of one-millimetre accuracy for the freeboard."

Comment: Table 5: It would be better if these data can be presented in a plot/plots (i.e., vertical profiles), which can show clearly the vertical structures and temporal variations of EC,pH, and Chl-a.

Response: Thanks for your suggestion. We presented the data in Table 5 in vertical plots which show clearly the vertical structures and temporal variations of EC, pH, and Chl $a$. Also, we made a short and clear explanation in the text for Fig.6 as follows: The vertical profiles of EC, pH and Chl $a$ show that EC was larger near the snow-ice surface than in congelation ice in the early melting stage, but the difference was no more obvious after April 14. pH was always smaller in snow-ice than in congelation ice. Chl $a$ content was less than 0.6 μg L$^{-1}$ with the maximum at the snow-ice – congelation ice interface.

Comment: L322: how did the data of EC on April 14 confirm the deposition of acidic substances from the atmosphere? And why did EC of ice increase after a snowfall?

Response: Thank you very much, there was a mistake. We meant pH for the acidic deposit and even that was not a proof. Therefore, we wrote "Atmospheric deposition of acidic substances was judged as the background for the low pH of snow-ice." There was a little increase of EC of congelation because of the melting of the snow-ice and the meltwater drainage down. We have rewritten section 3.3 (3.3 Ice geochemistry) in the manuscript and given a clear explanation.

Comment: L333-334: "Algae can grow in a slush layer within snow-ice, but not in consolidated ice because of lack of liquid water for living organisms." However, Fig.

5b shows clearly the Chl-a content in congelation ice increases gradually as melting proceeds. The increase in Chl-a content within the ice is likely to result from the increasing solar radiation and/or the decreasing surface albedo rather than the thinning ice cover.

Response: Yes, you are right, the increase in Chl $a$ content within the ice resulted from the increasing solar radiation and the formation of pores with liquid water in the congelation ice layer. This part in the manuscript was somewhat confusing and unclear. We revised the sentence by "Algae can grow under ice and in slush layers in ice all winter at sufficient photon flux conditions. In springtime algae growth occurs also in pores in ice containing liquid water. Chl $a$ content was similar in under-ice water and in ice until April 26, but then it increased in water and became much higher than in ice at the time of ice breakup, but still well below the first summer peak."

Comment: (3): it would be better if you present briefly the physical meaning of each formula.

Response: Thanks for your suggestion. We have done what was requested. Considering also your following comments and the comments of Reviewer 1#, we have revised heat budget text. The daily meteorological data from FMI was used to analyze the ice melting in the heat budget. The albedo was parameterized as $\alpha = 0.7$ for snow, 0.5 for fry ice and 0.3 for wet ice. After the revised input data for the heat budget model, the results of model simulation are in good agreement with the observed results.

Comment: L359: 4 W m$^{-2}$ cannot be a half of the incoming solar radiation in May.

Response: As said, $k_0(t)$ is not solar radiation but depends on the net radiation that includes both net solar and net longwave radiation. It is the part of the heat balance independent of the surface temperature, and the $k_1$ term then gives the correction due to the difference between air temperature and surface temperature (see the given reference). In Lake Pääjärvi site, $k_0 > 0$ in summer and $k_0 < 0$ in winter.

Comment: L363: does the bottom melting depend on the solar radiation?

Response: Yes it does, since sunlight heats under-ice water from where a part of the gained heat goes to ice melting. Indeed, Eq. (6) consists of the background heat flux from the deep water and a fraction $c$ of the solar radiation penetrated to the water:

$$Q_w = Q_{w0} + c(1 - \alpha)\gamma e^{-\lambda h}Q_{s0} \ , \tag{6}$$

where $\alpha$ is albedo, $\gamma$ represents the fraction of solar radiation that penetrates the surface, and $\lambda$ is the light attenuation coefficient.

Comment: L367: what do you mean by " $\gamma$ represents the fraction of light in solar radiation"? I guess it is the fraction that penetrates through the ice surface.

Response: Yes, you are right, we didn't make it clear for readers. We have revised " $\gamma \approx 0.5$ represents the fraction of light in solar radiation" by "$\gamma$ represents the fraction of solar radiation that penetrates the surface (the light band)"

Comment: L368: by Qs0=150W m$^{-2}$, is it the daily-averaged value? It looks like a daytimeaveraged value in April. If this is true, Qs0≈75 W m$^{-2}$, the melt rate can be 0.16 cm d$^{-1}$,close the observed rate of 0.18 cm d$^{-1}$.

Response: Yes, you are right, it's the averaged value in April. The FMI solar radiation data used now gives the average incident radiation in April as 184 W m$^{-2}$. E.g., take albedo as $\alpha = 0.5$, $\gamma = 0.5$ and $\lambda = 0.5$ m$^{-1}$, and then the resulting absorption by ice is $Q_A = (1 - \alpha)\gamma(1 - e^{-\lambda h})Q_{s0} \approx 5 - 10$ W m$^{-2}$ depending on the thickness $h$ that corresponds to the observed melt rates.

Comment: (6): Could please give a brief physical background of this equation? What do the terms mean at the right-hand side?

Response: Yes. We have added a brief physical background of equation (6): "and the bottom melting in Eq. (6) consists of the background heat flux from the deep water $(Q_{w0})$ and a part of the under-ice solar radiation."

Comment: L389: the second "freezing" should be "melting"

Response: Here, the first "freezing" is "freezing days", the second "freezing" is "freezing dates", they are two different meanings. But in the revision of the manuscript, this sentence has been deleted.

Comment: L397: what do you mean by "the ice freezing days"? freezing duration, or ice-covered duration?

Response: Sorry we did not make it clear, it should be ice-covered duration. However, we have revised the structure of the manuscript, and this sentence has been deleted.

Comment: L433: after the net radiation becomes positive? The net radiation is always positive, I guess. And whether or not the surface, interior, and bottom melting take place depends on different conditions of heat balance.

Response: Thanks for your suggestion. Net radiation (solar + longwave) can be positive or negative. Anyway, we have revised this sentence as: Melting of ice begins after the heat balance becomes positive and takes place at the surface, interior and bottom depending on depending on the ice structure and fluxes.

Comment: L467: "Yang et al. (2012) modelled…too late", what do you mean?

Response: Sorry, it's language issues. It should be "The ice breakup date modelled by Yang et al. (2012) turned out to be 12 d late." However, we have revised the structure of the manuscript, and this sentence has been deleted.

Comment: L482: "Therefore, … on the structures.", what do you mean?

Response: Sorry, it's language issues. We did not make it clear. We have revised the paragraph as: The quality of ice decay is important to ice mechanics due to the loss of ice strength (Ashton, 1986; Leppäranta, 2015). There are two important consequences.

First, the bearing capacity of ice $(P)$ decreases. This quantity scales as $M \propto \sigma_f h^2$,

where $\sigma_f$ is the flexural strength, and during the melting period ice thickness and strength both decrease, thickness due to melting at the boundaries and strength due to internal melting. The positive albedo feedback in the melting process produces a patchy ice cover, and together with the unpredictable bearing capacity the ice cover becomes a severe safety issue. Secondly, the two-dimensional yield strength of a lake ice cover scales as $P \propto \sigma_c h / L$, where $\sigma_c$ is the compressive strength of ice and $L$ is the lake size length scale. With decreased thickness and strength, wind stress may lead to ice breakage and ice movement on shores, where damage can be caused to man-made structures since the ice strength is still finite.

Comment: L527: a surge of phytoplankton under ice may indicate a positive net production, which uses CO2 to produce oxygen and biomass, so why it results in an increase of CO2?

Response: Sorry, we did not give a clear explanation in the manuscript. The reason why $CO_2$ increased is that after the enhancement of biological activity, the total respiration in water is greater than photosynthesis, which leads to the increase of carbon dioxide content (Li, 2016). After we have revised the structure of the manuscript, and these sentences have been deleted.

Li, R. L.: Variations of Sea Ice and Sea Water Characteristics and Their Effects on Immune Enzyme Activity of Shellfish in Liaodong Bay, Ph.D. thesis, Dalian university of Technology, China, 130 pp., 2016.

Finally, according to the adjustment of the structure and content of this manuscript, the references of this manuscript are also adjusted accordingly.

---

## Author Response (AR2)

Dear editor,

Thank you very much for your letter regarding our manuscript entitled "A field study on ice melting and breakup in a boreal lake, Pääjärvi, in Finland". We are truly grateful to the comments and suggestions from you and the reviewers. The manuscript has been carefully revised. The point-by-point answers to the comments and suggestions were listed as below.

**Response to Reviewer 1# Comments**

The revised text underwent essential restructuring and shortening and presents the results in a much more concise and coherent way than the previous version. The value of the new findings is now clear: the detailed quantification of the late phase of lake ice melt in terms of relationship between the surface, bottom, and internal ice melting rates contributes towards better understanding of the global balance of seasonal freshwater ice. The new observations on the biogeochemical parameters within and underneath the ice cover add a valuable facet on the effect of the ice melt on physical-biological coupling in lakes.

Some minor remarks:

The new section explaining the heat balance calculations provides the necessary background to the field estimates and puts them into the general context. If set earlier in the textflow, before the geochemistry results, closer connected to the field data on ice melt rates···

Response: Thanks for your advice. We interchanged the order of 3.3 and 3.4. And we changed all the numbers of the figures in 3.3 and 3.4.

The value of Qw0 of 1 W/m^2 is rather arbitrary. While the conductive flux from water to ice is indeed small, it can still vary within an order of magnitude. A range of possible values should be mentioned here.

Response: Thanks for your suggestion. We adjusted this part and gave the range of possible values of $Q_{w0}$.

The background term $Q_{w0}$ is not known but for molecular conduction the scale is $Q_{w0} = k_w \, \partial T / \partial z \sim 1$ W m$^{-2}$, where $k_w = 0.56$ W m$^{-1}$ °C$^{-1}$ is thermal conductivity of water, and in general mid-winter data suggest that $Q_{w0} < 5$ W m$^{-2}$. With $Q_{w0} = 1$ W m$^{-2}$, we have $Q_w = 11.5$ W m$^{-2}$ and the corresponding melt rate at the ice bottom would be 0.33 cm d$^{-1}$.

Some details on estimates of k0 in Eq. 4 would be useful to follow the derivations without searching through the cited literature.

Response: Thanks for your suggestion. We added some details on estimates of $k_0$ in Eq. 4.

where $k_0$ is independent of the surface temperature but depends on time, and $k_1 \sim 15$ W m$^{-2}$ °C$^{-1}$. In exact terms, $k_0$ includes the radiation balance and latent heat flux at

$T_a = T_0$, and their correction for $T_a \neq T_0$ plus the sensible heat flux is included in $k_1$. In Lake Pääjärvi, $k_0$ varies between $-50$ W m$^{-2}$ and $115$ W m$^{-2}$ in summer (Leppäranta and Wen, 2022).

L64 "but, however": remove either "but" or "hovewer"
Response: Changed as suggested.

L92 "final results": remove "final"
Response: We have removed "final".

Fig. 4 "Sluash": correct spelling
Response: You are right. We have corrected the spelling of "sluash" as "slush".

L297 remove "Now"
Response: We have removed "Now".

L432 remove "very"
Response: We have removed "very".

L434-435: The entire sentence can be removed as repetition.
Response: We have removed the entire sentence.

**Response to Reviewer 2# Comments**

The manuscript has been improved significantly in terms of its language, structure, and presentation, but some minor technical modifications should be taken care before possible acceptance.
[1] L20: delete "during"
Response: We have deleted "during".

[2] L41: in and below
Response: Changed as suggested.

[3] L310: but depends mainly on···
Response: Changed as suggested.

[4] Section 4.1: this section seems quite isolated from the main contents of the ms and maybe can be skipped over, and none of your results was used to analyze or understand the ice phenology and its variability. You mentioned "Field data of lake ice are very important to examine and predict ice phenology", but how you examined or predicted?
Response: Thanks for your suggestion. We have adjusted this section but not remove. In this manuscript, ice decay was monitored from the start to the final breakup resulting with a full time-series of the evolution of ice thickness, structure, and

geochemical properties. However, the ice break up date is also an significant ice phenological parameter during ice melting period. Based on the ice break data from the Lammi Biological Station, we can see the interannual variations of ice breakup date in a boreal lake, Pääjärvi. We mentioned "Field data of lake ice are very important to examine and predict ice phenology", we examined it from the following references and we put the references in the manuscript. Which can be seen in Line 356-359.

George, G. D.: The impact of the North Atlantic Oscillation on the development of ice on Lake. Windermere, Climatic Change 81, 455–468, 2007.

Williams, G., Layman, K. L. and Stefan, H. G.: Dependence of lake ice covers on climatic, geographic and bathymetric variables, Cold Reg. Sci. Tech., 40, 145–164, 2004.

Stefan, H. G., and Fang, X.: Simulated climate change effects on ice and snow covers on lakes in a temperate region, Cold Reg. Sci. Tech., 25, 137–152, 1997.

[5] Sections 4.2 and 4.3: Since the authors compared their results from the studied lake with those from others and other lakes, e.g. in terms of melting rate, heat flux from water, and geochemistry, it would be better if you can tell why they are different or even similar. And most important, can we get some common points or understanding when intercomparing, like in a broader scope?

Response: Thanks for your advice. We adjusted this part and added some explanation about why the melting rate, heat flux from water, and geochemistry are different or even similar. And we got some common points or understanding when comparing with other research. Which can be seen in Line 385-387, Line 393-394, Line 406, and Line 436-437.

[6] L428-432: This part can be removed or significantly shortened.

Response: Thanks for your suggestion. We have significantly shorted this part based on your comments.

Ice season has a specific role in the local environment and human life, and it has an impact on the lake ecology far beyond the ice period. Research of lake ice has largely increased to evaluate the impact of the predicted climate change on the ice phenology and properties. During the melting period, it is difficult to do fieldwork due to the deterioration of the ice cover, and therefore only few data have been collected.